# Classifying Interactions in a Synthetic Bacterial Community Is Hindered by Inhibitory Growth Medium

Andrea R. Dos Santos,[a] Rita Di Martino,[a] Samuele E. A. Testa,[a] Sara Mitri[a,b]

[a]Department of Fundamental Microbiology, University of Lausanne, Lausanne, Switzerland
[b]Swiss Institute of Bioinformatics, Lausanne, Switzerland

**ABSTRACT** Predicting the fate of a microbial community and its member species relies on understanding the nature of their interactions. However, designing simple assays that distinguish between interaction types can be challenging. Here, we performed spent medium assays based on the predictions of a mathematical model to decipher the interactions among four bacterial species: *Agrobacterium tumefaciens*, *Comamonas testosteroni*, *Microbacterium saperdae*, and *Ochrobactrum anthropi*. While most experimental results matched model predictions, the behavior of *C. testosteroni* did not: its lag phase was reduced in the pure spent media of *A. tumefaciens* and *M. saperdae* but prolonged again when we replenished our growth medium. Further experiments showed that the growth medium actually delayed the growth of *C. testosteroni*, leading us to suspect that *A. tumefaciens* and *M. saperdae* could alleviate this inhibitory effect. There was, however, no evidence supporting such "cross-detoxification," and instead, we identified metabolites secreted by *A. tumefaciens* and *M. saperdae* that were then consumed or "cross-fed" by *C. testosteroni*, shortening its lag phase. Our results highlight that even simple, defined growth media can have inhibitory effects on some species and that such negative effects need to be included in our models. Based on this, we present new guidelines to correctly distinguish between different interaction types such as cross-detoxification and cross-feeding.

**IMPORTANCE** Communities of microbes colonize virtually every place on earth. Ultimately, we strive to predict and control how these communities behave, for example, if they reside in our guts and make us sick. But precise control is impossible unless we can identify exactly how their member species interact with one another. To find a systematic way to measure interactions, we started very simply with a small community of four bacterial species and carefully designed experiments based on a mathematical model. This first attempt accurately mapped out interactions for all species except one. By digging deeper, we understood that our method failed for that species as it was suffering in the growth medium that we chose. A revised model that considered that growth media can be harmful could then make more accurate predictions. What we have learned with these four species can now be applied to decipher interactions in larger communities.

**KEYWORDS** cross-feeding, detoxification, consumer-resource model, spent media, experimental design

As they grow, microbes modify their environment. This affects other organisms living in their proximity, resulting in "indirect" or "environmentally mediated" interactions (1, 2). How to classify microbial interactions has been a subject of some debate, but broadly, they can be cooperative, competitive, or neutral based on one species' positive, negative, or absent effects on another species' growth, respectively (1).

The way in which such positive and negative effects are physically and chemically mediated may affect the survival of the interacting species (their ecology) and how selection acts on their traits (their evolution) (1–4). Positive interactions occur when one species improves the environment of another, either by reducing its adverse

Address correspondence to Sara Mitri, sara.mitri@unil.ch.

The authors declare no conflict of interest.

effects or by producing compounds that enhance the other's growth (1). Whether these improvements also benefit the acting species and/or are costly can affect evolutionary dynamics. For example, siderophores or nutrient-degrading enzymes are useful to their producers as well as other species but are quite costly (5). Nonproducing mutants can then invade the population of producers and destabilize the interaction. But positive interactions can remain stable over time if they are not exploitable: a species may take up nutrients that alter the pH to another species' benefit (6, 7) or secrete costless metabolic by-products, which can be cross-fed by other coinhabiting species (8, 9).

Predicting the long-term fate of competitive interactions is equally mechanism dependent. Competition can be due to one species enhancing harmful conditions (e.g., the production of bacteriocins) or removing beneficial ones (e.g., competition for nutrients) (1, 10). Under the latter, known as "exploitative competition," species compete for resources, and we expect them to evolve to occupy separate niches and compete less (11–14). Under more direct "interference competition," the production of toxins, antibiotics, or phage-like particles may result in arms races and species extinctions (10, 15). Competitive interactions often rely on direct cell-to-cell contact (15). In sum, even if positive and competitive interactions are easily measurable at the population level, understanding the mechanisms underlying these measured effects can change the predictions of long-term dynamics or environmental changes.

In natural communities, interactions occur simultaneously among many species, with little evidence of which molecule was produced or consumed by which species and which species it affects in which way. Identifying interactions and their molecular mechanisms in such complex webs is clearly quite challenging but can be achieved in a high-throughput manner using spent medium (SM) assays, which we show can distinguish between interaction types without needing to distinguish interacting species, e.g., by fluorescently labeling them. To develop and test the utility of these assays, small synthetic microbial ecosystems of up to a few dozen species are more practical (16–20): interspecies interactions are easier to disentangle and control, especially since the chemistry of the environment can be designed and community members can be genetically engineered or selected to exhibit specific interactions (21–26). Their simplicity also allows parameter estimations in mathematical models to predict community dynamics (24, 27–30).

Here, we aimed to decipher the interactions in a synthetic community that we studied previously, composed of four bacterial species, *Agrobacterium tumefaciens*, *Comamonas testosteroni*, *Microbacterium saperdae*, and *Ochrobactrum anthropi*, that can grow and degrade industrial machine oils (31). This community was dominated by positive interactions when we compared their growth in mono- and cocultures in the oil. However, the chemical complexity of the growth medium and the use of cocultures made it difficult for us to understand the mechanisms behind these positive interactions. Here, we sought to provide a more controlled environment and used a defined minimal medium (MM) to study the mechanisms behind the interactions among the four species. As we were interested in chemical and metabolic interactions that do not require cell-to-cell contact, we grew each species in the SM of all of the other species using an experimental design that allows us to distinguish between interaction types (e.g., exploitative versus interference competition). Ideally, these simple assays would suffice to identify all types of pairwise interactions without the need for detailed chemical analyses of secreted and consumed molecules.

We found two strong positive interactions mirroring our previous work: the pure SM of *A. tumefaciens* and *M. saperdae* shortened the lag phase of *C. testosteroni*. However, we were surprised that this positive effect was lost and even reversed if the spent media were replenished with the original growth medium (MM). Further investigation revealed that the no-carbon (NC) compounds (which cannot be used as carbon sources) in the replenished SM delay *C. testosteroni*'s growth despite the presence of enough available carbon sources and the SM. We then wondered whether *A. tumefaciens* and *M. saperdae* might remove the inhibitory

compounds from the environment for *C. testosteroni*, allowing it to grow sooner. Although such cross-detoxification seemed to be the most parsimonious explanation, we found no evidence to support it. Instead, using untargeted liquid chromatography-mass spectrometry (LC-MS), we identified at least three molecules secreted by *A. tumefaciens* and/or *M. saperdae* that could be metabolized by *C. testosteroni* and that shortened its lag phase. Our findings show that pinpointing the nature of positive interactions can be quite challenging, because growth media can sometimes be inhibitory and because inhibitory effects can be alleviated by either cross-detoxification or cross-feeding, but that spent medium assays and growth curve measurements can nevertheless be sufficient to distinguish between cross-feeding and cross-detoxification.

## RESULTS

*A. tumefaciens* **and** *O. anthropi* **responded to other species according to model predictions.** To identify the interactions among the four species in our simplified medium, we conducted spent medium (SM) assays by growing each species alone until stationary phase in a minimal medium (MM) containing a no-carbon (NC) part (salts and trace metals) and two carbon sources (CSs), glucose and citric acid (MM = NC+CS) (see Materials and Methods). We then removed the cells by filtration and grew each species in all other resulting SM. We compared the growth in SM to the growth in a medium comprising only the NC compounds, a positive control of MM, and two other conditions (Fig. 1A, left). First, we diluted (1:1) the SM in 2× NC so that the species have access to at least the same concentrations of salts and trace metals as those in the fresh MM (SM/2+NC), and second, SM was diluted (1:1) in 2× MM to contain at least the original concentrations of carbon sources, salts, and trace metals (SM/2+MM).

Our expectations for these five different experimental conditions (NC, MM, SM/2+NC, SM, and SM/2+MM) were established using a mathematical model (see Text S1 in the supplemental material) where we simulated the outcomes of six interaction types: exploitative competition, exploitative competition with interference competition, exploitative competition with cross-feeding, niche separation, niche separation with interference competition, and niche separation with cross-feeding (Fig. S1A and Fig. S2). We calculated the area under each simulated growth curve (AUC) and compared it to those of the negative and positive controls (NC medium and MM) (Fig. 1A). For example, if two species compete for the same carbon source or if a species is grown in its own SM (exploitative competition), the carbon source in the model is depleted such that growth in SM/2+NC and SM is identical to growth in NC medium. The replenishment of the carbon sources in SM/2+MM restores growth to the level of the positive-control MM. The remaining base expectations are shown in Fig. 1A and Fig. S2.

As in the model, the spent medium experiments were analyzed by calculating the area under the $OD_{600}$ (optical density at 600 nm) growth curve (AUC). This measure is well suited to quantifying interactions in batch cultures (31), where the importance of growth rate, lag duration, and final yield can vary depending on the culture duration. The AUC combines them without the need for complicated parameter fitting, and we examine individual growth curves in detail to better analyze interesting outcomes.

The growth of *A. tumefaciens* and *O. anthropi* could be classified according to the six anticipated scenarios (Fig. 1B; Fig. S3) with the help of a decision tree that we constructed based on the predictions of the model (see Fig. S1B). We have omitted data on *M. saperdae* as quantifying its growth was problematic due to contradictions between our measurements (Fig. S3A and Fig. S4A and B). Surprisingly, the growth patterns of *C. testosteroni* did not correspond to any of the expected scenarios (Fig. 1B).

We first verified the behavior of *A. tumefaciens* and *O. anthropi* by quantifying which carbon sources the four species use (see Materials and Methods). We found that *A. tumefaciens* consumes mostly glucose, *C. testosteroni* consumes mostly citric acid, and *O. anthropi* consumes both, while *M. saperdae* consumes little of either (Fig. 1C). In agreement with this, *A. tumefaciens* reflects the niche separation model in the spent media of *C. testosteroni* (with some evidence of cross-feeding) and *M. saperdae*. Since *A. tumefaciens* does not consume citric acid but overlaps with *O. anthropi* in consuming

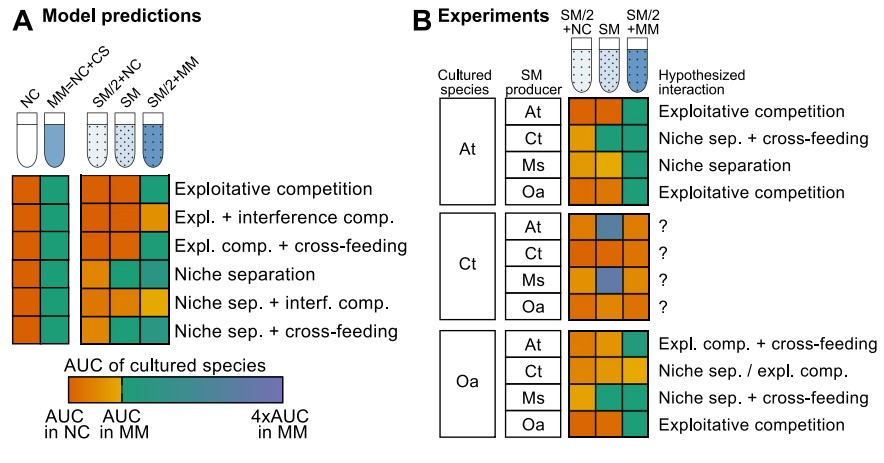

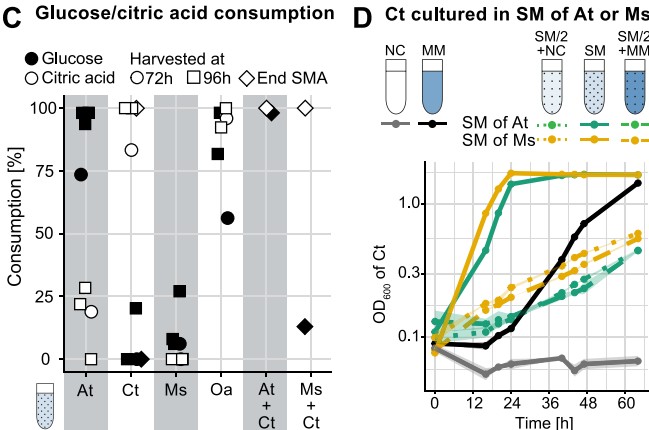

**FIG 1** Spent medium assays. (A) We used our model to simulate the growth of species under different types of positive or negative interactions, as illustrated in Fig. S1A in the supplemental material. Five simulated growth conditions allowed us to distinguish among these four different interaction types: no-carbon (NC) medium containing only salts and trace metals but no carbon sources, minimal medium (MM) composed of NC medium with 15 mM glucose and 10 mM citric acid (CS, carbon sources), a mix of the spent medium of a given partner species (50%) and NC medium (50% of a 2× concentrated solution) (SM/2+NC), the pure spent medium of a partner species (SM), and a mix of the spent medium of a partner species (50%) and minimal medium with carbon sources (50% of a 2× concentrated solution) (SM/2+MM). Square colors reflect the area under the growth curve (AUC) under these different conditions relative to the AUC in NC medium and MM (gradient key). See Fig. S2 for the model growth curves that generated these. (B) Results of spent medium assays, where each row shows how a given species grew under the different SM conditions (SM/2+NC, SM, and SM/2+MM) for each SM producer normalized by their growth in NC medium and MM (see Fig. S3 for growth curves and AUCs with statistical comparisons). Comparing these patterns to the predictions in panel A (with the help of the decision tree in Fig. S1B) determined the hypothesized interactions. When *C. testosteroni* (Ct) was grown in the spent media of others, interactions did not qualitatively match any predicted scenarios. The growth of *M. saperdae* (Ms) is not shown here, as the OD and the CFU data were not consistent. (C) Consumption (percent) of glucose and citric acid by the 4 species based on commercial chemical kits applied to the spent media in stationary phase (72 and 96 h of growth). We also analyzed the SM of *C. testosteroni* after growth in the pure SM of *A. tumefaciens* (At) and *M. saperdae* (samples collected at the end of the SM assay [SMA] after ~64 h, thus labeled "end SMA"). Oa, *O. anthropi*. (D) Growth curves of *C. testosteroni* as OD$_{600}$ values on a log$_{10}$ scale over time when grown under the five culture conditions, with either *A. tumefaciens* or *M. saperdae* generating the spent media ($n = 3$) (± standard deviations [SD] [error bars are present but very small]). Colors indicate spent medium-producing species.

glucose, the SM interactions of both *A. tumefaciens* and *O. anthropi* exhibit some exploitative competition. Similarly, in the SM of *C. testosteroni*, *O. anthropi* follows a pattern in between exploitative competition and niche separation. We observed some evidence of cross-feeding from *A. tumefaciens* and *M. saperdae* to *O. anthropi* and interference competition from *C. testosteroni* to *O. anthropi* (see Fig. S3B for statistics), but we do not explore this further. Instead, we focus on why *C. testosteroni* did not fit our model's predictions.

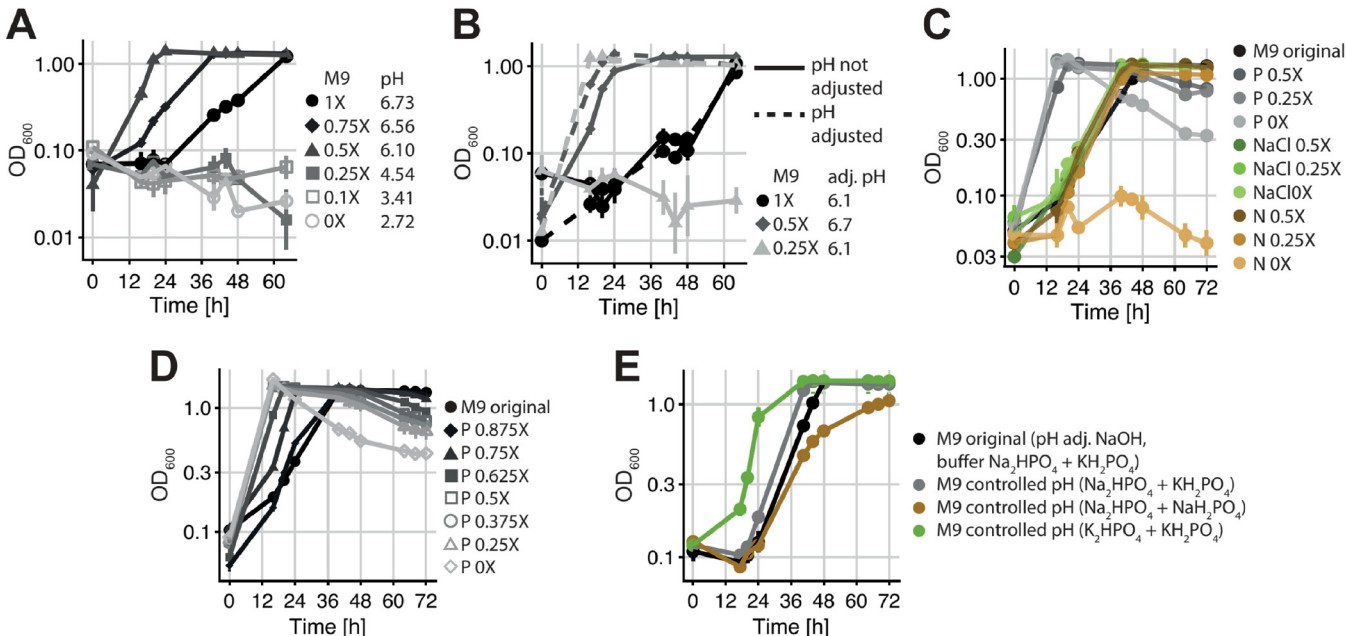

**FIG 2** Growth of *C. testosteroni* (as OD$_{600}$ values on a log$_{10}$ scale) in minimal medium (MM) made with no-carbon (NC) medium where we modified the M9 part. (A) Growth in MM with decreasing concentrations of M9. The resulting pH is indicated for each fresh MM, showing that decreasing the concentration of M9 also decreases the pH. (B) Growth in MM with decreasing concentrations of M9 with and without manual pH adjustment to either pH 6.1 or 6.7 using NaOH. Decreasing the concentration of M9 while keeping the pH close to neutrality shortens the lag phase of *C. testosteroni*. (C) Growth in MM with M9 that varies in the concentrations of its main components: phosphate (overall Na$_2$HPO$_4$ plus KH$_2$PO$_4$ concentration [abbreviated P]), sodium chloride (NaCl), and nitrogen (NH$_4$Cl [abbreviated N]). The pH is adjusted to 6.7 with NaOH under all conditions (the pH of the original MM). The overall concentration of phosphate seems to be the sole factor that affects *C. testosteroni*'s lag phase. (D) Growth in MM with M9 varying in its overall concentration of phosphate (Na$_2$HPO$_4$ plus KH$_2$PO$_4$ [abbreviated P]). The pH is adjusted to 6.7 with NaOH under all conditions (the pH of the original MM). (E) Growth in MM with M9 that has phosphate compounds comprising either only Na$^+$ ions (Na$_2$HPO$_4$ plus NaH$_2$PO$_4$), only K$^+$ ions (K$_2$HPO$_4$ plus KH$_2$PO$_4$), or both ions (Na$_2$HPO$_4$ plus KH$_2$PO$_4$) but with adjusted ratios so that the pH is 6.7 ("controlled pH") with no NaOH adjustment. For all graphs, the means ± SD are plotted (*n* = 3).

***C. testosteroni* has a shorter lag phase in the pure spent media of *A. tumefaciens* and *M. saperdae* but grows poorly under all other conditions.** When *C. testosteroni* grows in the pure SM of either *A. tumefaciens* or *M. saperdae*, its AUC is significantly higher (both df = 5 [*P* < 0.001] [by a *t* test with Bonferroni correction]) than that in MM, which is due to a much shorter lag phase (Fig. 1D). However, when we replenished the growth medium (SM/2+MM), *C. testosteroni* grew significantly worse than it did in the original MM medium (both df = 5 [*P* < 0.001]) (Fig. 1B and D). We observe the same pattern in its own SM and *O. anthropi*'s SM (growth in SM/2+MM is significantly worse than that in MM [*P* < 0.001]) (Fig. S3B). This is surprising because SM/2+MM should contain at least the same concentration of carbon sources as that in MM. This led us to suspect that some of the replenished compounds might impair the growth of *C. testosteroni* because they end up at higher concentrations than those in the original medium. Accordingly, we tested a first hypothesis, that one or more of the compounds in the NC medium inhibit the growth of *C. testosteroni*, while *A. tumefaciens* and *M. saperdae* can reduce their concentration and thereby shorten the lag phase of *C. testosteroni*. According to our model, such "cross-detoxification" could be a valid explanation for the observed patterns (Fig. S5).

**No-carbon compounds delay the growth of *C. testosteroni* but are not reduced by *A. tumefaciens* or *M. saperdae*.** To explore whether the NC medium could affect the growth of *C. testosteroni*, we first manipulated its two main ingredients, M9 minimal medium and Hutner's vitamin-free mineral base (HMB) (see Materials and Methods). While reducing the concentration of HMB had no positive effect (Fig. S4C and D), a small decrease in the concentration of M9 shortened the lag phase of *C. testosteroni*, while reducing it further had a detrimental effect (Fig. 2A; see Fig. S4E for CFU). As changing the concentration of M9 changes both the pH and osmolarity simultaneously, we next tested the effect of each ingredient separately and found that both (i) increasing the pH and (ii) decreasing the M9 concentration but keeping the pH constant (lowering the osmolarity)

shortened the lag phase of *C. testosteroni* (Fig. 2B). To assess if specific compounds in M9 (see Materials and Methods) could influence the lag phase of *C. testosteroni*, we tested the effect of each compound on the growth of *C. testosteroni* by decreasing its concentration in the NC (Fig. 2C). Of all of the compounds, we found that the total concentration of phosphate ($Na_2HPO_4$ and $KH_2PO_4$) was the only one that influenced the lag phase of *C. testosteroni* independently of the pH (Fig. 2D). Changing the ratio of the two ions ($Na^+$ and $K^+$) by changing the ratio of the corresponding salts ($Na_2HPO_4$ and $KH_2PO_4$) also affected the lag phase: a higher proportion of potassium shortened it, while a higher proportion of sodium lengthened it (Fig. 2E).

These findings suggest that the NC medium is suboptimal for *C. testosteroni* and lengthens its lag phase: a controlled pH, a lower phosphate concentration, or a smaller sodium-to-potassium ratio in M9 allows *C. testosteroni* to grow earlier. Accordingly, our most parsimonious explanation for its SM behavior (Fig. 1B) was that *A. tumefaciens* and *M. saperdae* can modify at least one of these factors in the medium. However, when we measured pH, osmolarity, and the concentrations of phosphate, sodium ions, and potassium ions in the SM of the 4 species (Fig. S6), we found that none of them differed between the original MM and the SM of *A. tumefaciens* and *M. saperdae* in a way that would explain the shortened lag phase of *C. testosteroni*.

In sum, several properties of the NC medium lengthen the lag phase of *C. testosteroni*, but *A. tumefaciens* and *M. saperdae* appear to be unable to significantly modify these properties in a way that would explain why *C. testosteroni* grows so well in their SM. We therefore rejected our first hypothesis.

***C. testosteroni* feeds on metabolites produced by *A. tumefaciens* and *M. saperdae*.** Another hypothesis that could explain the shortened lag phase of *C. testosteroni* in the SM of *A. tumefaciens* and *M. saperdae* is that their SM contains metabolic by-products that allow *C. testosteroni* to grow earlier. To find such candidate molecules, we performed an untargeted liquid chromatography-mass spectrometry (LC-MS) analysis (see Materials and Methods) on the SM of *A. tumefaciens*, *C. testosteroni*, and *M. saperdae* and on the SM of *A. tumefaciens* and *M. saperdae* after *C. testosteroni* had grown in it to assess if *A. tumefaciens* and *M. saperdae* secrete molecules that *C. testosteroni* then consumes (Fig. 3A; see Fig. S7 for the full list).

We identified several compounds that follow this pattern (Fig. 3A) and selected three, based on availability, cost, and ease of use, to assess their effects on the lag phase of *C. testosteroni*: hypoxanthine, oxoglutarate, and proline (Fig. 3B). As LC-MS yielded only the relative abundances of each compound, we added several concentrations to *C. testosteroni* growing in MM. We found that a range of concentrations shortens the lag phase of *C. testosteroni* significantly (oxoglutarate, 0.39 mM to 12.5 mM; proline, ≥0.39 mM; hypoxanthine, ≥0.62 mM [complete statistical results are available in reference 32]). At high-enough concentrations, all three metabolites also increased the final yield of *C. testosteroni*, suggesting that they act as carbon sources (oxoglutarate, 0.09 mM to 12.5 mM; proline, ≥0.78 mM; hypoxanthine, ≥1.25 mM [complete statistical results are available in reference 32]). At its two highest concentrations (25 and 50 mM), oxoglutarate even had an inhibitory effect on *C. testosteroni*. To confirm that these three metabolites could act as carbon sources, we cultured *C. testosteroni* in NC medium containing each of the metabolites at an intermediate concentration (oxoglutarate and proline, 1.56 mM; hypoxanthine, 1.25 mM) as the sole carbon source and observed significant growth in all three cases (Fig. 3C).

It appears, then, that these three metabolites are being cross-fed by *C. testosteroni*. If they are responsible for the shortening of the lag phase of *C. testosteroni* in the SM of *A. tumefaciens* and *M. saperdae*, the replenishment of the medium should cancel this effect, as we observed in the spent medium assays (SM/2+NC and SM/2+MM) (Fig. 1B and D). We grew *C. testosteroni* in MM supplemented with each of the three metabolites (at concentrations based on the data in Fig. 3B) and added increasing concentrations of the NC medium (Fig. 3D). As in our original experiments, the growth of *C. testosteroni* was impaired despite the presence of the metabolites and the original carbon sources under all conditions compared to the positive controls (no addition of NC compounds or metabolites),

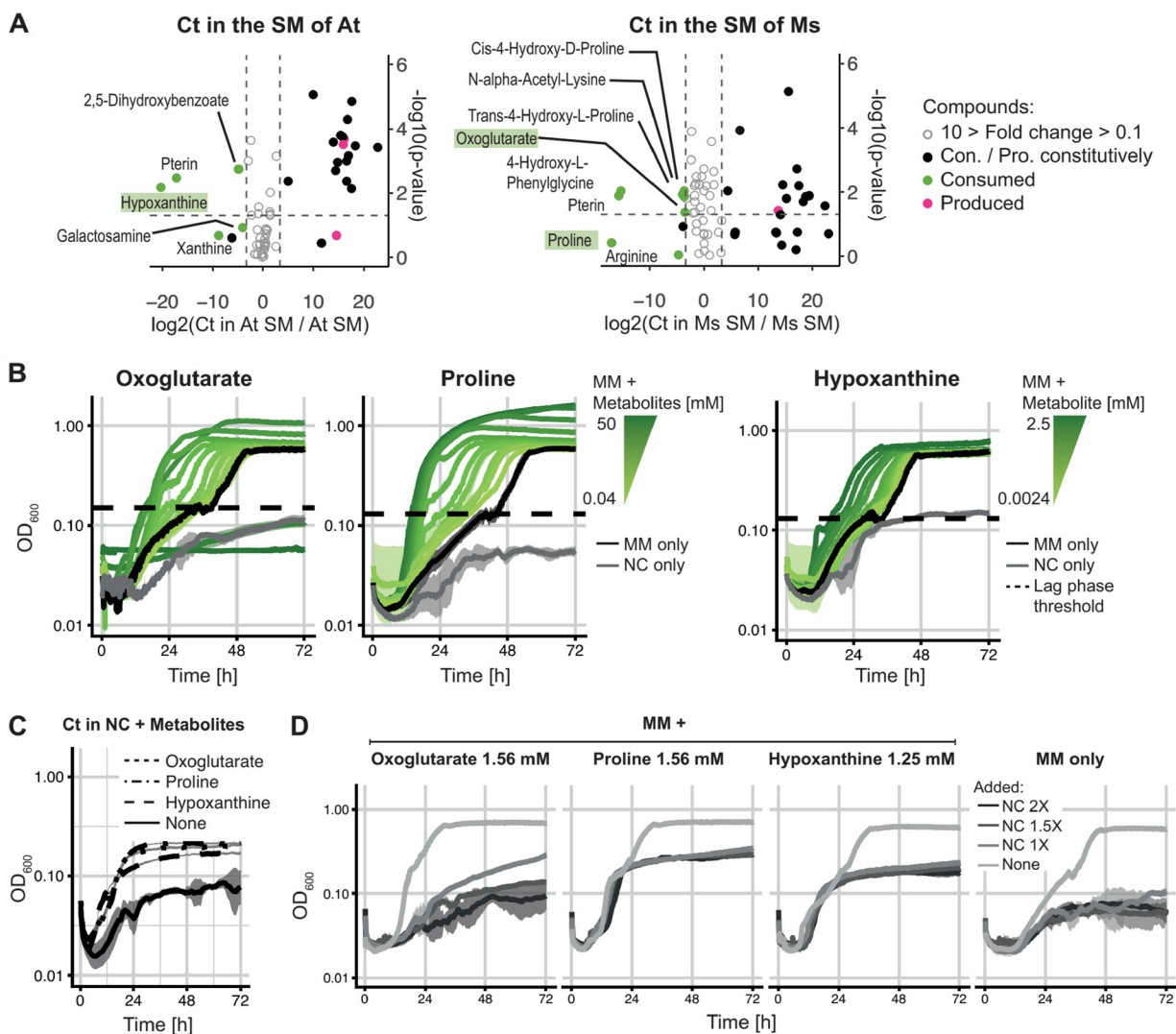

**FIG 3** Identification of three cross-fed metabolites (oxoglutarate, proline, and hypoxanthine) and their effect on *C. testosteroni*. (A) Untargeted metabolomics analysis (focusing on polar molecules) performed on the SM of *A. tumefaciens*, *C. testosteroni*, and *M. saperdae* and on the SM of *A. tumefaciens* and *M. saperdae* after *C. testosteroni* grew in them to identify metabolites that were produced by *A. tumefaciens* and *M. saperdae* to then be consumed by *C. testosteroni* (relative quantification). (Left) The x axis shows the log$_2$ ratios (fold changes) between the abundance of each metabolite in the spent medium of *A. tumefaciens* after the growth of *C. testosteroni* to the abundance of each metabolite in the spent medium of *A. tumefaciens*. If metabolites are on the left of the graph (negative), *C. testosteroni* consumed them from the *A. tumefaciens* SM, while if they are on the right (positive), *C. testosteroni* produced them. Gray metabolites between the dashed lines have a fold change of <10, which we did not consider. The y axis shows the log$_{10}$ P values; the significance threshold of a P value of 0.05 is represented by the dashed line. Metabolites in green boxes were chosen for further analysis. (Right) Same as the left panel but for the SM of *M. saperdae* (n = 3 for both panels). One-factor analysis of variance (ANOVA) (on log$_{10}$-transformed data) was used to test the significance of metabolite changes under the different conditions. (B) Effects of oxoglutarate, proline, and hypoxanthine on the growth of *C. testosteroni* at a range of concentrations added to MM, measured as the OD$_{600}$ on a log$_{10}$ scale over 72 h. The means are plotted, and the transparent areas around the curves represent the standard deviations. As a control, *C. testosteroni* was grown in MM or NC medium (n = 4). (C) Effect of 1.56 mM oxoglutarate, 1.56 mM proline, and 1.25 mM hypoxanthine on the growth of *C. testosteroni* in NC medium only, with NC medium alone as a control, over 72 h. The OD was measured every 10 min; the means are plotted, and the transparent areas around the curves represent the standard deviations (n = 3). The scale of the y axis is smaller than that in panel D to better show the growth curves. (D) Effects of intermediate concentrations of each metabolite in MM with increasing concentrations of NC medium (50% replenishment in either 1×, 1.5×, or 2× NC medium), measured as the OD$_{600}$ on a log$_{10}$ scale over 72 h. The means are plotted, and the transparent areas around the curves represent the standard deviations. As a control, *C. testosteroni* was grown in MM (under the same conditions [n = 3]). The data show that all three metabolites could act as carbon sources and could shorten *C. testosteroni*'s lag phase, and the effect was reversed upon the addition of NC medium.

supporting our hypothesis. These results are in line with our updated model (Fig. S5), which includes the inhibitory effect of the medium.

Overall, we find that different properties of the NC compounds can lengthen the time that *C. testosteroni* takes to start growing. Metabolites secreted by *A. tumefaciens* and *M. saperdae*, at least the three that we tested, can reduce this effect through cross-

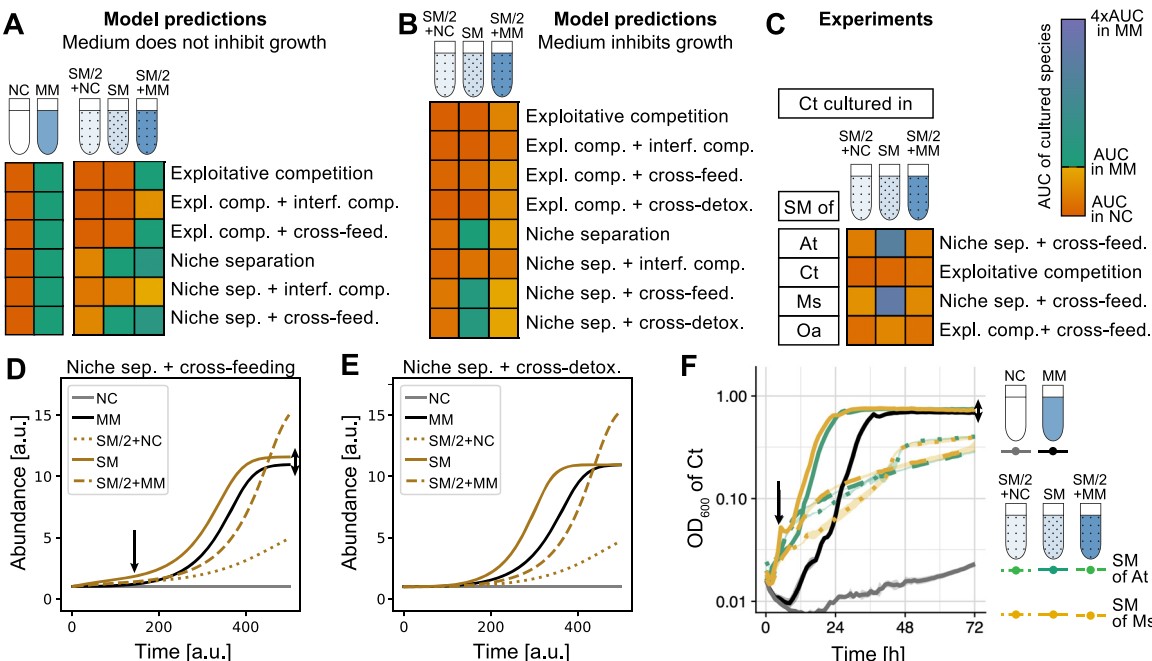

**FIG 4** The revised model with inhibitory growth medium predicts *C. testosteroni*'s growth patterns better than the original model. (A to C) Predictions of the original model (same as in Fig. 1A) (A), predictions of the updated model where the environment inhibits growth (B), and data for the growth of *C. testosteroni* in the SM of the four different species (C). Although the predictions do not match quantitatively, the model in panel B qualitatively matches the experimental data in panel C. (D and E) Predictions for niche separation plus cross-feeding (D) and niche separation plus cross-detoxification (E) in the updated model (inhibitory growth medium). In panel D, the single arrow shows the growth bump in SM compared to MM at around 120 arbitrary units (a.u.) due to a switch in carbon sources, and the double arrow shows the higher final yield in SM. The features highlighted with the arrows are absent in panel E, where the bump is missing and the final yields are identical in MM and SM. (F) These model predictions can be compared to a repetition of the experiment shown in Fig. 1D in a 96-well plate with higher time resolution. As it shows a small bump in the two species' SM (solid yellow and green lines [see the left arrow]), indicating a switch in the carbon source, and a higher final yield than for growth in fresh medium (MM) (black [see the right arrow]), we conclude that just by observing the growth curves at a high resolution, we can correctly classify the interaction as niche separation plus cross-feeding (matching panel D). Note that the predictions in panel D still do not correspond perfectly to the experimental data, indicating that there may be additional effects at play.

feeding. However, when the concentration of NC compounds is high enough, the effect of the metabolites is insufficient, and *C. testosteroni* grows very little on the time-scale of our experiments.

## DISCUSSION

Before running our SM assays, we used a simple mathematical model to generate our base expectations under different experimental conditions. We were surprised when the behavior of one species, *C. testosteroni*, did not correspond to the predictions of any of the scenarios simulated by the model. Our further analysis revealed that this was because our model assumed that the growth medium could either allow cells to grow or not, but it could not inhibit species, such that increasing its concentration would delay their growth. In hindsight, we know that designing growth media for different species can be quite challenging (33, 34), and bacterial responses to stress have been extensively studied. It should come as no surprise, then, that some media have negative effects on some bacterial species.

Based on this new intuition, we updated our model to cover cases where the medium inhibits growth (Fig. 4B; see also Fig. S5 in the supplemental material). The patterns in this new model fit qualitatively with what we observed for *C. testosteroni* (Fig. 4C) but also drew our attention to another issue: it was challenging to distinguish between cross-feeding and cross-detoxification. Even though cross-detoxification seemed the more parsimonious explanation, all of our experiments led us to reject it as the underlying interaction. The new model does, however, give two clues to distinguish these two types of positive interactions without chemical analyses: first, in the pure spent medium (SM), the final yield in a cross-

detoxification interaction should not exceed that of the original growth medium (under some simplifying assumptions), while this should be the case for cross-feeding (Fig. 4D, right double arrow; Fig. S5), and second, in cross-feeding, one may detect a first short stationary phase as cells switch their metabolism to consume the second carbon source (Fig. 4D, left arrow). We reran our original experiment in a plate reader to obtain higher time resolution, and indeed, we found a significant difference in the final yields of *C. testosteroni* in MM and the pure SM of *A. tumefaciens* ($P < 0.01$) and *M. saperdae* ($P < 0.001$) as well as a small "bump" at the beginning of the SM growth curves (Fig. 4F, arrows) (complete statistical results on the final yield and the length of the lag phase are available in reference 32). These two features, obtained by high-resolution growth curve measurements until stationary phase, make it possible to distinguish between cross-feeding and cross-detoxification without the need for further molecular analyses.

Our simple spent medium assays and updated model can be used to classify the dominant interactions, at least in a defined medium with few carbon sources. Spent medium assays are more powerful than direct coculture experiments, where the directionality of interactions is obscured. Our model is not, however, meant to cover all possible scenarios, as we constructed it to help decipher the results of our own experiments. For example, we consider interference competition only as an increase in lag-phase-lengthening compounds and do not model killing, which may be common in multispecies cocultures. Another assumption of our model is that each species consumes a single one of the supplied carbon sources, regardless of how many carbon sources are present. Finally, the model is meant to be a qualitative reference, and classifications were done based on a decision tree (Fig. S1B) that compares the five spent medium conditions to one another. A sweep of parameters related to the production and consumption of metabolites is provided in Fig. S8.

Even in the simple scenario that we have studied, we may be missing additional layers of interactions among these four species. First, all three SM metabolites that we tested turned out to be cross-fed, suggesting that others may have a similar effect. This aligns with previous research suggesting that cross-feeding might be quite common (35, 36), particularly for species that are in a suboptimal environment that contains few carbon sources (37, 38) or is toxic (31). It may be, then, that the difficulty in finding a growth medium that can sustain the individual growth of all members of a community is precisely what allows us to observe cross-feeding interactions. In fact, positive interactions are often "accidental," resulting from the secretion of costless metabolic by-products by the few species that can grow (8, 19, 38–45). While the secretion of metabolites like amino acids by bacteria might seem counterintuitive, several mechanisms such as the maintenance of cell homeostasis (the release of overproduced metabolites) or cell lysis can explain the secretion of costly metabolic by-products (46). We were also surprised that *M. saperdae* produced several metabolites (Fig. 3A) and affected *C. testosteroni*'s growth, even though its own population size did not increase significantly (Fig. S4A and B). This aligns with the results of other studies showing that cross-feeding does not require bacterial growth (8). Indeed, the absence of growth does not necessarily indicate metabolic inactivity, as metabolic activity is required to produce enough energy for survival. This suggests that in larger bacterial communities, such as the gut microbiome, non- or slow-growing species should not be ignored, as they may still significantly affect other community members.

Despite our efforts, it remains unclear why the minimal medium delays *C. testosteroni*'s growth and why the cross-fed compounds allowed it to start growing sooner. One hypothesis is that *C. testosteroni* experiences osmotic stress in the minimal medium, which can be reflected in the length of its lag phase (47, 48). Given that the other 3 species seemed robust to this stress, the metabolites that they secrete could, once consumed, help *C. testosteroni* to cope with this stress. Proline, for example, can act as a "compatible solute" (49–52), which is a molecule that bacteria synthesize or take up to balance osmotic pressure in hyperosmotic environments. Alternatively, the metabolites could act as metabolic precursors, allowing *C. testosteroni* to synthesize its own compatible solutes *de novo*. This may be the case for oxoglutarate, which is a direct intermediate in the Krebs cycle (53). In *Escherichia coli*, oxoglutarate is taken up from the environment (54) but is also leaked (55), hinting that it may be

involved in extracellular exchange. Similarly, the purine derivative hypoxanthine is an important nitrogen source and participates in nucleic acid synthesis via the pentose phosphate salvage pathway (56). Interestingly, hypoxanthine was found to mediate interactions influencing biofilm formation between *Bacillus subtilis* and soil bacteria whose cell-free supernatants were analyzed similarly to our approach (high-performance liquid chromatography [HPLC], nuclear magnetic resonance [NMR] spectroscopy, and high-resolution mass spectrometry [HR-MS]) (57). Another hypothesis is that *C. testosteroni* requires a metabolic shift to grow on citric acid compared to other carbon sources and that this increases its lag phase due to a high enzymatic cost (58, 59). The presence of metabolites secreted by *A. tumefaciens* and *M. saperdae* could then allow it to metabolize citric acid more rapidly. Distinguishing between these different hypotheses could be achieved by engineering *C. testosteroni* to report on osmotic stress, by isotopically labeling the carbon sources and monitoring their metabolic by-products from *A. tumefaciens* and *M. saperdae* that are later consumed by *C. testosteroni*, and/or by testing the role of the remaining identified metabolites.

The ability to classify interspecies interactions to the level of distinguishing cross-feeding from cross-detoxification, for example, is not just a matter of curiosity but is key to understanding and predicting community dynamics (2, 60, 61). Even mechanistic details of cross-feeding can affect community dynamics. La Sarre et al. (60) showed that increasing the concentration of a cross-fed metabolite can render it toxic to the partner species, leading to a new community equilibrium. To make matters even more complicated, each species pair is likely to interact in more than one way, but the effects that we observe are cumulative (61). Here, we showed how *M. saperdae*, for example, produced a whole series of compounds and that *C. testosteroni* could feed on the two compounds that we tested. But it may well be that other compounds have small inhibitory effects on *C. testosteroni* and that changing the environmental conditions could increase their production or leakage rates and alter community dynamics (60).

In conclusion, our work proposes that carefully designed spent medium assays together with a simple mathematical model can help to map out the dominant metabolic interactions in more detail than simply labeling them as positive or negative. High-resolution growth curve measurements can even help to distinguish cross-feeding from cross-detoxification. We have showcased this using a small, synthetic bacterial community in a defined medium, in which we could verify the readouts from the growth curves with more detailed analyses. It remains to be seen whether our approach would scale up to more high-throughput approaches in larger communities (e.g., as in reference 38) and more complex environments. But ultimately, such simple experimental approaches are needed to predict the dynamics of natural microbial communities.

## MATERIALS AND METHODS

**Cell culture preparation.** Species were grown in monoculture in TSB (tryptic soy broth) overnight (28°C with shaking at 200 rpm), diluted to an $OD_{600}$ of 0.05 in fresh TSB, and incubated again for 3 h in order to reach exponential growth. Each culture was then washed 2 times in PBS (phosphate-buffered saline) (centrifugation for 15 min at 4,000 rpm at room temperature), and the final bacterial pellets were resuspended in adequate medium so that the initial $OD_{600}$ would be 0.1. The compositions of the different media used are available elsewhere (32).

**CFU measurement.** To measure the CFU, we sampled 20 $\mu$L of our cultures, diluted them in 180 $\mu$L of PBS in 96-well plates, and proceeded with 10-fold dilutions down to $10^{-7}$. The dilutions were plated onto TSA (tryptic soy agar) as drops and then spread into lines.

**Spent medium assays.** Spent media (SM) from each of the four species were obtained by growing them in large volumes ($V = 30$ mL) in MM until they reached stationary phase (~72 h to 96 h, decided by $OD_{600}$ determinations). After approximately 72 h, we measured the $OD_{600}$ and determined if all of the cultures were in stationary phase (known OD values). If they were in stationary phase, we harvested all cultures. If not all of the cultures were in stationary phase, we waited until approximately 96 h (the next day) and harvested all cultures. We proceeded in this way because (i) the SM assays would take a significant amount of time to start and (ii) we wanted to avoid storing SM samples in the fridge. We then centrifuged the bacterial culture (20 min at 4,000 rpm at room temperature) and collected the supernatants. We centrifuged the supernatants again to eliminate as many bacterial cells and as much debris as possible before filtering them using vacuum filters (TPP vacuum filtration "rapid"-Filtermax, polyethersulfone [PES] membrane, 0.22 $\mu$m). From the SM, we prepared 3 medium conditions to test the effect of the SM on our 4 species: SM/2+NC (SM:2× NC medium, 1:1), pure SM only, and SM/2+MM (SM:2× MM, 1:1). Our control conditions were fresh NC medium (negative control) and fresh MM (positive control). We

grew the 4 species in monocultures under these 5 conditions ($V$ = 4 mL) over 60 h, measured the $OD_{600}$ over time (using an Ultrospec 10 cell density meter; Biochrom), and performed CFU counts before the initial incubation, at 24 h and 48 h. We calculated the area under the $OD_{600}$ growth curve (AUC) and used this value as a proxy for growth (DescTools [62]; R version 4.1.2). We repeated these SM assays in 96-well plates ($V$ = 200 $\mu$L) to increase the resolution of our data and measured the $OD_{600}$ every 10 min for 72 h using a microplate reader (BioTek Synergy H1 at 28°C with continuous double-orbital shaking) (see reference 32 for the full data set).

**Glucose, citric acid, phosphate, sodium, potassium, and osmolarity quantification.** To quantify glucose, citric acid, phosphate, sodium, and potassium in the SM of *A. tumefaciens*, *C. testosteroni*, and *M. saperdae*, we used different chemical kits. We generated SM for each species as described above (see the section on spent medium assays, above) (total incubation time of ~89 h) according to the specific protocols of the following kits to determine which concentration of SM to test given the theoretical concentrations of the tested compounds in fresh MM: a glucose (HK) assay kit (catalog number GAHK-20; Sigma), a citric acid kit (catalog number K-CITR; Megazyme), a phosphate assay kit (colorimetric) (catalog number ab65622; Abcam), a sodium assay kit (colorimetric) (catalog number MAK247; Sigma), and a potassium assay kit (fluorometric) (catalog number ab252904; Abcam). Osmolarity was measured in each SM sample using an osmometer (Osmomat 030; Gonotec) (the full data set is available in reference 32).

**Metabolomics analyses of SM samples.** Untargeted metabolomics analyses were performed for us at the Metabolomics Platform, Faculty of Biology and Medicine, University of Lausanne, on the following samples, focusing on polar (water-soluble) compounds: fresh minimal medium (MM); SM of *A. tumefaciens*, *C. testosteroni*, and *M. saperdae* (generated as described above for the spent medium assays); and SM of *A. tumefaciens* and *M. saperdae* after *C. testosteroni* grew in them (total incubation time of ~60 h). To summarize the procedure, we first identified which metabolites were produced by *A. tumefaciens*, *C. testosteroni*, and *M. saperdae* when grown in monoculture in MM (fresh MM compared to the SM of *A. tumefaciens*, *C. testosteroni*, and *M. saperdae*) and then compared this list of metabolites to the ones identified in the SM of *A. tumefaciens* and *M. saperdae* after *C. testosteroni* grew in them (the SM of *A. tumefaciens* and *M. saperdae* compared to the SM of *A. tumefaciens* and *M. saperdae* after *C. testosteroni* grew in them). Using these comparisons, we could identify 64 compounds that were produced by *A. tumefaciens* and *M. saperdae* and then consumed by *C. testosteroni* or that were absent in the SM of *A. tumefaciens* and *M. saperdae* but were later produced by *C. testosteroni*. From this list, we considered only the compounds consumed by *C. testosteroni* with a fold change of at least 10. For further details, see Text S1 in the supplemental material (32).

**Testing the effect of the metabolites oxoglutarate, proline, and hypoxanthine on *C. testosteroni*.** We performed similar protocols for all of the compounds. For oxoglutarate and proline, the same concentration was tested. In a 96-well plate, we added 180 $\mu$L of water in wells B(1) to E(1) (4 replicates). We then added 20 $\mu$L of a 1 M stock solution of either oxoglutarate or proline so that these wells contained 100 mM of the metabolite tested. The other wells in lines B to E were filled with 100 $\mu$L of water. Serial dilutions from wells B(1) to E(1) to B(11) to E(11) were performed by transferring 100 $\mu$L each time (2×). In this way, we obtained 11 concentrations to test on *C. testosteroni* (from 50 mM to 0.04 mM), in 4 replicates. To these wells, we then added 100 $\mu$L of *C. testosteroni* cultures that were in 2× MM at an OD of 0.2 (according to the method described above for cell culture preparations). In this way, the final concentration of MM is 1×, and the final OD is 0.1 (as usually tested). As controls, we grew *C. testosteroni* in MM (in 4 wells, we mixed 100 $\mu$L of water with 100 $\mu$L of the *C. testosteroni* culture [in 2× MM at an OD of 0.2]) in addition to *C. testosteroni* in NC medium (in 4 wells, we mixed 100 $\mu$L of water with 100 $\mu$L of the *C. testosteroni* culture [in 2× NC medium at an OD of 0.2]). For hypoxanthine, we proceeded slightly differently as its solubility is much lower than those of the two other metabolites. We prepared a hypoxanthine stock at 5 mM and directly added 200 $\mu$L to wells B(1) to E(1). We then followed the same logic as that for oxoglutarate and proline. We could test concentrations from 2.5 mM to 0.002 mM. Growth was assessed by measuring the $OD_{600}$ every 10 min for 72 h using a microplate reader (BioTek Synergy H1 at 28°C with continuous double-orbital shaking). The effects of each metabolite on the lag phase of *C. testosteroni* and on its final yield compared to both parameters in MM (positive controls) were assessed statistically (see reference 32 for complete statistical results) (R version 4.1.2).

**Testing oxoglutarate, proline, and hypoxanthine as carbon sources.** To test if the metabolites alone could support the growth of *C. testosteroni*, we grew *C. testosteroni* in NC medium containing intermediate concentrations of the metabolites (oxoglutarate and proline at 1.56 mM and hypoxanthine at 1.25 mM) (Fig. 3D). We prepared *C. testosteroni* cultures (according to the method described above for cell culture preparations) at an OD of 0.2 in water, and in a 96-well plate, we mixed 100 $\mu$L of the culture with 100 $\mu$L of 2× NC medium plus the metabolites. Growth was assessed by measuring the $OD_{600}$ every 10 min for 72 h using a microplate reader (BioTek Synergy H1 at 28°C with continuous double-orbital shaking).

**Testing the effects of oxoglutarate, proline, and hypoxanthine with increasing concentrations of NC compounds on *C. testosteroni*.** To test the effects of the metabolites on *C. testosteroni* when increasing concentrations of NC compounds are added, we chose an intermediate concentration of each metabolite (oxoglutarate and proline at 1.56 mM and hypoxanthine at 1.25 mM). We tested the growth of *C. testosteroni* in MM plus metabolites when 2×, 1.5×, or 1× NC medium was added. We thus prepared *C. testosteroni* cultures in 2× MM plus metabolites at an OD of 0.2 and one culture of *C. testosteroni* in 2× MM only (as a control). In a 96-well plate, we added 100 $\mu$L of the *C. testosteroni* culture to 100 $\mu$L of either 2×, 1.5×, or 1× NC medium or water (as a control). In this way, the MM is 1× concentrated, and *C. testosteroni* is at an OD of 0.1 (as usually tested). As a negative control, we also grew *C. testosteroni* in NC medium (OD = 0.1). Growth was assessed by measuring the $OD_{600}$ every 10 min for 72 h

using a microplate reader (BioTek Synergy H1 at 28°C with continuous double-orbital shaking) (see reference 32 for the full data set for the above-described assays).

**Mathematical model.** The mathematical model is described in Text S1, parameters are as in Table S1, and the code can be found at https://c4science.ch/diffusion/12144/repository/master/.

## SUPPLEMENTAL MATERIAL

Supplemental material is available online only.
**TEXT S1**, PDF file, 0.2 MB.
**FIG S1**, PDF file, 0.5 MB.
**FIG S2**, PDF file, 0.2 MB.
**FIG S3**, PDF file, 1.4 MB.
**FIG S4**, PDF file, 1.9 MB.
**FIG S5**, PDF file, 0.2 MB.
**FIG S6**, PDF file, 0.4 MB.
**FIG S7**, PDF file, 0.6 MB.
**FIG S8**, PDF file, 0.2 MB.
**TABLE S1**, PDF file, 0.1 MB.

## ACKNOWLEDGMENTS

We thank Christoph Keel, Rizlan Bernier-Latmani, Björn Vessman, Shota Shibasaki, Aurore Picot, Afra Salazar, and three anonymous reviewers for very useful and constructive feedback on the manuscript. Untargeted metabolomic profiling was performed at the Metabolomics Platform, Faculty of Biology and Medicine, University of Lausanne. We acknowledge the entire team for their work from sample preparation and data acquisition to data processing and metabolite identification. We particularly thank Julijana Ivanisevic and Hector Gallart-Ayala for discussing the analyses with us. We also thank Alice Wallef and Nastassia Quévit for additional experiments and Gwenaël Labouebe for access to the osmometer. We thank Björn Vessman for developing the first version of the mathematical model code, available at https://gitlab.com/eccemic/facilitation2019.

A.R.D.S. is funded by Swiss National Science Foundation grant PCEGP3_181272, R.D.M. is funded by European Research Council grant 715097, S.E.A.T. is funded by NCCR Microbiomes, and S.M. is funded by all three grants.

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
