## [Reviewer comments · mSystems]

Classifying interactions in a synthetic bacterial community is hindered by inhibitory growth medium

Andrea Dos Santos, Rita Di Martino, Samuele Testa, and Sara Mitri

Corresponding Author(s): Sara Mitri, University of Lausanne

Review Timeline:

Submission Date:	March 10, 2022
Editorial Decision:	April 12, 2022
Revision Received:	July 2, 2022
Editorial Decision:	August 2, 2022
Revision Received:	August 27, 2022
Accepted:	September 15, 2022

Editor: Karoline Faust

Reviewer(s): The reviewers have opted to remain anonymous.

Transaction Report:

DOI: <https://doi.org/10.1128/msystems.00239-22>

April 12, 2022

Dr. Sara Mitri
University of Lausanne
Lausanne
Switzerland

Re: mSystems00239-22 (Classifying interactions in a synthetic bacterial community is hindered by inhibitory growth medium)

Dear Dr. Mitri:

I am pleased to inform you that the reviewers judged this work to be interesting and well implemented, so in principle, we expect to accept it for publication in mSystems. However, there are first a number of issues to address. More specifically, a discussion is needed concerning the usage of the area under the growth curve as opposed to more classical measures of interaction strength and concerning the role of the model. The wider implications of the findings also need to be discussed (do we need to explicitly model an inhibitor in the environment or is it sufficient to simply capture its effect via a lower growth rate).

Below you will find instructions from the mSystems editorial office and comments generated during the review.

Preparing Revision Guidelines

Sincerely,

Karoline Faust

Editor, mSystems

Journals Department
Reviewer comments:

Reviewer #1 (Comments for the Author):

In their manuscript "Classifying interactions in a synthetic bacterial community is hindered by inhibitory growth medium" the authors show that even rather standard media can have an inhibitory effect on bacteria which can obscure underlying (in this case positive) interactions.

I find the work interesting. First, because it addresses the topic of detoxification as a potential mode of interactions. And second, because it shows how complex chemical processes underlying microbial interactions can be. I feel both points are rather understudied in general. The experiments are well done and many controls and alternatives have been tested. So it feels like the authors did experimentally what they could do. Nevertheless I have a few issues concerning the interpretation and presentation of data.

My major concerns are:

1) The authors use area under the curve (AUC) of OD measurements as a metric for growth and in the end for interaction. Since the AUC is impacted by the lag time the authors find an impact on the "interaction" by the inorganic compounds of the media even when the final number of cells at steady state does not change. However, I have a hard time to call this really a change in interaction since the steady state value of population densities does not change and thus the overall fitness is not impacted. Or in other words, the way interaction is defined here is different from what for example co-culture experiments or standard ecological models (like Lotka Volterra) would do and this change in definition seems crucial to "observe" the stated effects. I think the authors should clarify this point.

2) The authors used replenished spent media to test if interactions are driven by resource competition or "interference competition". However, upon adding fresh media back into the spent media they also dilute the media by quite a lot (2fold). How can in this case the effect of diluting the media be separated from replenishing substances. E.g. how can a growth inhibition after adding back fresh media to spent media caused by inhibiting media distinguished from a dilution of cross-fed resources?

3) The authors use the predictions of a model to classify the observed outcomes. However, like every model also this model has parameters that are set by the authors to specific values (as described in the supplement). I have the feeling that the outcome of the model should depend on the choice of those parameters at least quantitatively and maybe also qualitatively. For example in Fig 1A row 5 the outcomes in condition 4 and 5 should depend on how strong NS is compared to IC? So how robust are the outcomes concerning the parameter choice? In the light of this, it is also not clear to me how the different scenarios can be classified as in Fig 1B. How is decided whether an experimental outcome is in agreement with an theoretical prediction given the impact of the parameters on this classification? Does this classification change with the choice of parameters?

Minor point:

For me the way the data is presented is quite hard to follow. It seems the story is presented in a chronologically way and every hypothesis and their falsification are spelled out. This makes it at least for me quite confusing. I feel the paper could be way more focused on the important parts and structured in a way that follows a more logical "flow". A few things to consider:

- Why is all the data shown in Fig2B,C when it is stated in the text, that most of the data is ignored in the following?
- There are so many abbreviations that I quickly lost track which abbreviation means what. Especially in Fig1A there is a whole list to define the abbreviations just to use the abbreviations directly next to it. This does not save any space. Along the same line I had always to look back to remember what conditions 1-5 are. Fig2 all the dark shades in A, B and D I cannot tell apart.
- I am not sure what the model adds here? I think the results of Ct/At and Ct/Ms pairs are surprising even without a model?
- The color and linestyle choices are often hard to decipher. E.g. in Fig 1D there are 3 different colors for conditions 1-5 but only two of them appear in the plot. The third color (green-yellow) stands for different cases. It took me some time to get it.

I don't want to force the authors to write their text in a certain way. So above points should be more seen as suggestions, but I feel the data could be presented much clearer which would also make it more appealing for a broader audience.

Reviewer #2 (Comments for the Author):

Review on:

Classifying interactions in a synthetic bacterial community is hindered by inhibitory growth medium

I find the idea behind the paper interesting which is to explore indirect interactions in the synthetic bacterial community using a spent media approach. The authors used a mathematical model and four species synthetic bacterial communities to study the indirect interaction between them. The findings are interesting and especially since the authors found that the growth media toxicity can be alleviated by competitors. I think the study adds a valuable understanding of microbial community assembly. I especially value well-thought-out interaction types and connecting that to the model and experimental results (essentially ideas in Fig1).

Some minor comments for improving the manuscript

1)
Lines 464 "they reached stationary phase (~72h to 96 hours, decided by OD600 determination)."

How did you choose when to stop the growth and prepare the filtrate? Could this timing be critical?

2)
Keywords: You use the term "Consumer resource model" to me this is usually predator-prey or host-parasite models. Please check.

3)
Figure 1 has a lot of abbreviations. I understand why, but would there be any options to make the figure easier for the reader?

4)
Line 60 "more aggressive "interference competition"," I am not sure if interference competition is more aggressive, it's just more direct / different than resource competition? I see what you mean, but wording it this way gives the impression that interference competition is stronger.

5)
Pekkonen & Laakso did some work with the filtrate idea many years ago. Please check if these papers are worth citing. E.g.

"Pekkonen, Minna, Tarmo Ketola, and Jouni T. Laakso. "Resource availability and competition shape the evolution of survival and growth ability in a bacterial community." PLoS One 8.9 (2013): e76471."

6)
Please check the Fig S1 legend "??"

Reviewer #3 (Comments for the Author):

In this paper, Dos Santos et al. attempt to classify interactions between four species in a synthetic microbial community by growing each species in the spent media of each other species. However, they find that the outcomes of these measurements do not conform with the expectations of a model. In particular, they find that one species (Ct) has a pattern of interactions such that it grows best in the pure spent media of two of the other species, and slightly worse when the spent media is supplemented by more of the original base media. They determine that this is due to cross-feeding of particular metabolites by Ct.

The work appears thorough and carefully done, and the paper is well-written overall (with a few key areas to improve clarity; see below). However, I am struggling to see how their results have general significance for the field, rather than being just a quirk of their particular system. To the extent the authors claim a general significance for these results, I find their arguments misleading.

The authors interpret their results as being an important example of how "cross-feeding could alleviate the negative effects of a

challenging environment" (lines 123-124), but I find this claim ill-defined. The essence of the claim appears to be that the authors believe there is a fundamental difference between the "benign environment" scenarios in the first row of Fig. S1, and the "inhibitory environment" scenarios in the second row. The difference in every case is just the presence of the inhibitor C3, but how is the presence of this inhibitor any different from simply assigning a slower growth rate to species S2 in the environment? If species S1 actively removed C3, then I can see how it plays a dynamic role in the different interaction architectures, but as is, it just seems like a fixed part of the background medium that doesn't need any explicit modeling (outside of the $r_{i,j}$ parameters in Eq. 1). Related to this, the authors use the word "inhibitory" for this effect (e.g., in the abstract on line 19), but my point is that inhibition must be relative to something, and it's not clear what, since the authors are only considering a single medium type.

Related to this, I didn't understand the significance of the inhibitory components of M9 to the later discovery that Ct was consuming metabolites produced by At and Ms. It doesn't seem very surprising that they can alter various components of the medium and change the lag time of a population, but I don't see what this has to do with the cross-feeding story that arises later. (The authors claim that the "minimal medium delays Ct's growth" (lines 278-279), but again, delay relative to what?) To me, the point here isn't that M9 delays growth, but simply that the metabolites from At and Ms accelerate growth relative to the base media. The inhibitory effects of the M9 components seem completely irrelevant to this, outside of the authors' speculation that the cross-fed metabolites might alleviate this stress (lines 384-385, but there is no evidence for this).

Line-by-line comments

- Lines 87-91: Here the authors introduce their system, but I think they need to provide a little more background on where this system comes from and why it is interesting beyond just citing their previous work.
- Lines 126-127: The title of this subsection sounds too vague. What are the "others"? What does "following anticipated scenarios" mean? I would suggest something like "At and Oa respond to other species according to model-predicted classifications."
- Lines 144-159: The link between the experimental media conditions (conditions I-V) and the theoretical interaction classes (EC, EC+IC, etc.) was confusing. Eventually I discovered Fig. S12 that seems to be the precise definition of the correspondence (is this what they use to construct Figs. 1A,B?), but this wasn't mentioned until the Discussion. The authors need to explain these details more precisely in this section, because they are crucial for everything that follows.
- Line 145: There should be a reference around here to the mathematical model details in Supplementary Note S3.
- Fig. 1D: The lines here are hard to distinguish. My understanding is that for conditions III-V there are supposed to be two different colors for each line style, but the legend confuses this because it also shows these lines in a third color (brown) which doesn't seem to actually be used in the plot. I think it would be best to just enumerate each line style and color combination separately in the legend, plus use a better set of line styles (the line styles for conditions III and V are too similar).
- Fig. 1D: Why don't the authors use the log-scale version of this plot (Fig. S11) here instead? Their claim is about lag time, but lag time differences are practically impossible to separate from growth rate differences on linear-scale plots, so the log-scale is really essential here. In any case, whichever alternative version of this plot ends up in the supplement should be cited in the caption and accompanying main text.
- Fig. 2: If the point of this figure is to show how the lag times differ across media conditions, why didn't the authors plot exactly that? I do appreciate the authors' inclusion of so many raw growth curves, so that readers can see the raw data as much as possible, but for the main text I think the paper needs more summary figures that more directly show the important trends (e.g., in lag time specifically, not necessarily the whole growth curve). Distinguishing the growth curves was especially hard in these plots, since the color schemes are not so obvious and some trends are even non-monotonic (as in Fig. 2A). I would certainly still want to include the growth curves themselves, but they could be supplementary figures. (It would also be more helpful to plot these growth curves on log-scales if the key feature is lag time differences.) This comment applies to Fig. 3B as well.
- Line 361: There is a glitch in one of the citations here.
- Fig. S1 caption: There is a glitch in a reference to what probably should be Supplementary Note S3.
- Fig. S4: The significance markers don't make sense in some of the cases here. Aren't they testing for significant differences of cases III-V compared to case II (set to 1)? Some data that almost exactly matches case II (e.g., At grown on At, case V) is still shown as significant. Shouldn't that be not significant?
- Fig. S5: Why don't the authors directly plot OD vs. CFUs/mL, rather than plotting each separately vs. time? That would show the correlation (or not) more clearly, especially since the axes for the two here are on completely different scales (OD is linear, CFUs/mL is log, so comparing them is very hard).
- Tables S1-S8: These tables contain information about the comparisons of growth curve traits (lag times and final yields), but

these quantities themselves are not plotted anywhere (related to aforementioned comment on Figs. 2 and 3B). These plots seem critical.

- Tables S1-S8: Also, the supplement would be easier to navigate if the tables were all separate from the figures, rather than intermixed.

- Fig. S11: There is an arrow in this figure that I believe is explained by the caption to Fig. 4, but it should be explained in this figure's caption as well.

- Eq. 1 (Supplementary Note S3): In Eq. 1b, ρ_i should not be written as a function of C_k , because the index k is summed over in the denominator of the lag time factor. This dependence should also be removed in Eqs. 1a and 1c, where it suggests that the growth rate of species i depends on pairs of compounds, whereas it only depends on each compound separately with a global lag time factor. Also, the authors should cite previous work that developed this model, or explain which parts of the model are original to this paper.

Response to referees

Manuscript number: mSystems00239-22

Title: "Classifying interactions in a synthetic bacterial community is hindered by inhibitory growth medium"

We are very grateful to all reviewers for their comments, which we think have helped to clarify and improve the manuscript. Reviewers' comments are written in black below, and our responses in blue font and indented text. We hope that you will find that our updated manuscript now merits publication.

Reviewer #1 (Comments for the Author):

In their manuscript "Classifying interactions in a synthetic bacterial community is hindered by inhibitory growth medium" the authors show that even rather standard media can have an inhibitory effect on bacteria which can obscure underlying (in this case positive) interactions.

I find the work interesting. First, because it addresses the topic of detoxification as a potential mode of interactions. And second, because it shows how complex chemical processes underlying microbial interactions can be. I feel both points are rather understudied in general. The experiments are well done and many controls and alternatives have been tested. So it feels like the authors did experimentally what they could do. Nevertheless, I have a few issues concerning the interpretation and presentation of data.

We are very glad that the reviewer appreciated our work and agrees with us on the value of studying detoxification and cross-feeding and the chemical processes that drive positive interactions.

My major concerns are:

1) The authors use area under the curve (AUC) of OD measurements as a metric for growth and in the end for interaction. Since the AUC is impacted by the lag time the authors find an impact on the "interaction" by the inorganic compounds of the media even when the final number of cells at steady state does not change. However, I have a hard time to call this really a change in interaction since the steady state value of population densities does not change and thus the overall fitness is not impacted. Or in other words, the way interaction is defined here is different from what for example co-culture experiments or standard ecological models (like Lotka Volterra) would do and this change in definition seems crucial to "observe" the stated effects. I think the authors should clarify this point.

It is true that we do not define fitness based on the steady state population density, which is what Lotka Volterra (LV) models assume. This is because we are not using a chemostat system here, which is where you would achieve such a steady-state and where an LV model would be more appropriate. Instead, in a batch culture, the duration of the experiment will have an important effect on fitness, such that terminating it earlier would select for a faster growth rate or shorter lag phase rather than a larger final population size. Indeed, our initial experiments lasted just over 60h (Fig. 1D), by which point C_t in the base medium had not yet reached steady state. We only found out what the steady state was when we explicitly lengthened the experiment to test the prediction of the model (Fig. 4F). The AUC then summarizes growth rate, lag duration and final population size for a given batch culture length.

In any case, the reviewer is correct that all of this will not be intuitive to our readers and needs justification. We have added the following sentence after the first mention of AUC (L. 181ff):
“This measure is well-suited to quantifying interactions in batch culture (Piccardi et al, 2019), where the importance of growth rate, lag duration, and final yield can vary depending on culture duration. The AUC combines them without the need for complicated parameter fitting, and we examine individual growth curves in detail to better analyse interesting outcomes.”

Add a small section to the discussion about the choice of consumer-resource model and not LV?

2) The authors used replenished spent media to test if interactions are driven by resource competition or "interference competition". However, upon adding fresh media back into the spent media they also dilute the media by quite a lot (2fold). How can in this case the effect of diluting the media be separated from replenishing substances. E.g. how can a growth inhibition after adding back fresh media to spent media caused by inhibiting media distinguished from a dilution of cross-fed resources?

The problem the reviewer mentions is inherent to the design of spent media experiments. If we add anything to spent media, we will dilute it and with it, its effects, whether positive or negative. This is precisely the reason why we ran so many conditions: e.g. we test the undiluted spent medium, the replenishment with and without carbon sources. The reduction of the effect of the spent medium by dilution is not always easy to predict, hence the use of the mathematical model.

Regarding the reviewer's example (distinction between the dilution of cross-fed compounds or inhibiting medium in SM/2+NC), a reduced positive effect of cross-feeding in a benign environment would reduce the final yield compared to the pure SM (which should be higher than in MM), while replenishing with an inhibitory medium would increase the lag phase. The growth curves can help to make this distinction (Fig. S2 bottom row vs. S5 5th row), but the AUCs are sufficient in principle: if there is cross-feeding and niche separation, the AUC in the SM should be greater than if there is none. Another clue that the medium is inhibitory comes from a species' growth in its own spent medium. For Ct, its growth was impaired when the SM was replenished in MM or in NC medium. More generally, the decision tree shown in Fig. S1B can help where the outcomes are not intuitive.

3) The authors use the predictions of a model to classify the observed outcomes. However, like every model also this model has parameters that are set by the authors to specific values (as described in the supplement). I have the feeling that the outcome of the model should depend on the choice of those parameters at least quantitatively and maybe also qualitatively. For example in Fig 1A row 5 the outcomes in condition 4 and 5 should depend on how strong NS is compared to IC? So how robust are the outcomes concerning the parameter choice? In the light of this, it is also not clear to me how the different scenarios can be classified as in Fig 1B. How is decided whether an experimental outcome is in agreement with an theoretical prediction given the impact of the parameters on this classification? Does this classification change with the choice of parameters?

We agree that in the first version of the manuscript we did not conduct a proper parameter sweep. We have done so now (within reason: we change one parameter at a time and focused on the parameters related to the interactions, e.g. compound production or uptake

rates), and have included a new supplementary figure showing the effect of these parameter changes (Fig. S8). We found that increasing the uptake or production rate of inhibitory compounds (cross-detoxification and interference competition, respectively) only had quantitative effects on the outcome. We did find, however, that producing a higher concentration of a nutritious compound (cross-feeding) could change the results qualitatively as well. This helped us realize that our decision tree (Fig. S1B) was not accurate and we have now fixed that figure. In short, we find that if large amounts of the cross-feeding compound are produced, it is difficult to distinguish underlying niche separation vs. exploitative competition.

To answer the reviewer's second question on how the classification was made: we used the results of our model, together with the ground truth established through additional experiments including metabolomics to build the decision tree in Fig. S1B to classify interactions. These decisions all rely on comparing measurements relative to one another, e.g. the AUC in SM/2+NC versus in NC, or the final yield in SM versus MM.

This analysis (along with comments from other reviewers) helped us to realize that our model is not exhaustive. For example, we only consider interference competition that lengthens the lag phase, but if it were to have bactericidal effects, the decision tree might change. We therefore toned down the narrative and emphasize that the model was designed specifically as an aid to interpret our data in this paper. In the discussion section, we have now added the following text: "Our model is not, however, meant to cover all possible scenarios, as we constructed it to help decipher the results of our own experiments. For example, we only consider interference competition as an increase in lag-phase-lengthening compounds, and do not model killing, which may be common in multi-species co-cultures. Another assumption of our model is that each species consumes a single carbon source, regardless of how many are present. Finally, the model is meant as a qualitative reference and classifications were done based on a decision tree (Fig. S1B) that compares the five spent media conditions to one another."

Minor point:

For me the way the data is presented is quite hard to follow. It seems the story is presented in a chronological way and every hypothesis and their falsification are spelled out. This makes it at least for me quite confusing. I feel the paper could be way more focused on the important parts and structured in a way that follows a more logical "flow". A few things to consider:

We appreciate the reviewer's concern. It was indeed quite a difficult paper to write. Initially we tried to distill the message separately from the timeline in which we did the experiments, but found that it was often sounding artificial and dishonest, or omitting what we thought were interesting aspects of our findings. We revised these ideas again following the reviewer's comments, but still think that the current structure is the most straight-forward. We have nevertheless tried to streamline the writing more and to reduce the abbreviations in the main text to improve readability.

• Why is all the data shown in Fig2B,C when it is stated in the text, that most of the data is ignored in the following?

This figure makes the point that the carbon-free part of the medium is responsible for the long lag phase of Ct. We make this point in the title of the paper: the medium is inhibitory. Fig. 2 shows that this is because the M9 buffer is not optimized for this species. Even though these conditions are not altered by the other species (shown in supplement), they are what made the data interpretation challenging, which is the main point of the paper.

- There are so many abbreviations that I quickly lost track which abbreviation means what. Especially in Fig1A there is a whole list to define the abbreviations just to use the abbreviations directly next to it. This does not save any space. Along the same line I had always to look back to remember what conditions 1-5 are. Fig2 all the dark shades in A, B and D I cannot tell apart.

As this was pointed out by several reviewers, we have revised the figures accordingly. Now the only abbreviations are NC (no carbon), MM (minimal medium) and SM (spent medium). We think this has greatly improved readability, although the figures now contain more text. We have also changed the gradient colors to make the distinctions between values that lie below AUC in MM and ones that lie above.

The graphs in Fig. 2 have been modified so that in addition to the gradient of black-gray there are shapes instead of simple points so that each specific condition can be distinguished with no ambiguity.

- I am not sure what the model adds here? I think the results of Ct/At and Ct/Ms pairs are surprising even without a model?

The model is useful because it is quite complicated to navigate the outcomes of the different spent media experiments and conditions. This can be illustrated by the reviewer's question above as to how diluting the spent medium containing a cross-fed metabolite would result in a similar outcome as the medium being inhibitory. The model helps us to simulate these scenarios and show why the outcome would be different (as explained above). In other words, the model helped us to explore the parameter space and generate the decision tree in Fig. S1B. So even if the results of Ct/At and Ct/Ms are interesting in themselves, the model helped us to decide what hypotheses to test and how to interpret the results of our experiments.

- The color and linestyle choices are often hard to decipher. E.g. in Fig 1D there are 3 different colors for conditions 1-5 but only two of them appear in the plot. The third color (green-yellow) stands for different cases. It took me some time to get it.

We agree with the reviewer and have now improved the labelling of Fig. 1D to improve readability.

I don't want to force the authors to write their text in a certain way. So above points should be more seen as suggestions, but I feel the data could be presented much clearer which would also make it more appealing for a broader audience.

We thank the reviewer for pointing this out. These suggestions have indeed been very useful to help us see how confusing some of our plots were. Although we have not rewritten the text with a different perspective, we hope that the new figures and the reduction in acronyms has helped in interpreting the data.

Reviewer #2 (Comments for the Author):

I find the idea behind the paper interesting which is to explore indirect interactions in the synthetic bacterial community using a spent media approach. The authors used a mathematical model and four species synthetic bacterial communities to study the indirect interaction between them. The findings are interesting and especially since the authors found that the growth media toxicity can be alleviated by competitors. I think the study adds a valuable understanding of microbial community assembly. I especially value well-thought-out interaction types and connecting that to the model and experimental results (essentially ideas in Fig1).

We would like to thank the reviewer for these encouraging comments!

Some minor comments for improving the manuscript

- 1) Lines 464: "they reached stationary phase (~72h to 96 hours, decided by OD600 determination)."
How did you choose when to stop the growth and prepare the filtrate? Could this timing be critical?

We estimated that the cultures were in stationary phase when the OD stabilized. From growing these species many times we know more or less what OD they should reach when in stationary phase. However, this time was also influenced by experimental constraints. If on the day of the 72h mark the cultures were not yet completely in stationary phase in the morning (which could happen), we would not proceed with harvesting the spent media and starting the spent media assays on that day (even if only a few more hours were needed) as these experiments take a significant amount of time to set up. These choices were also motivated by our decision to always use spent media on the day they were harvested, so that we did not have to store them (fridge or freezer), which we worried might affect their composition (e.g. spontaneous degradation).

We also considered that harvesting time could affect the residual quantities of carbon, as well as the concentrations of certain secreted metabolites. However, we expect that those differences would be negligible, as the metabolism of bacteria in stationary phase is greatly reduced.

We have now expanded this in the methods section by adding the following sentences: "After approximately 72 hours, we measured the OD₆₀₀ and determined if all the cultures were in stationary phase (known OD values). If they were in stationary phase, we harvested all cultures. If not all of them were in stationary phase we waited until approximately 72 hours (the next day) and harvested all cultures. We proceeded this way because (i) the SM assays would take a significant amount of time to start and (ii) to avoid storing SM samples in the fridge."

- 2) Keywords: You use the term "Consumer resource model" to me this is usually predator-prey or host-parasite models. Please check.

Predator-prey models are often referred to as Lotka-Volterra models, where interactions between species are phenomenological (not modelled explicitly via metabolites). Instead, consumer-resource models include the substrates as variables in the models. A new review paper that we are involved in outlines these different approaches:
<https://www.nature.com/articles/s41559-022-01746-7>

3) Figure 1 has a lot of abbreviations. I understand why, but would there be any options to make the figure easier for the reader?

We have revised Fig. 1 and 4 to have fewer abbreviations. Now the only abbreviations are NC (no carbon), MM (minimal medium), CS (carbon sources) and SM (spent medium). We think this has greatly improved readability, although the figures now contain more text. We have also changed the gradient colors to make the distinctions between values that lie below AUC in MM and ones that lie above. We thank the reviewer for motivating us to improve this figure!

4) Line 60 "more aggressive "interference competition", I am not sure if interference competition is more aggressive, it's just more direct / different than resource competition? I see what you mean, but wording it this way gives the impression that interference competition is stronger.

Good point, the formulation was too anthropomorphic. We have changed it to "direct".

5) Pekkonen & Laakso did some work with the filtrate idea many years ago. Please check if these papers are worth citing. E.g. à "Pekkonen, Minna, Tarmo Ketola, and Jouni T. Laakso. "Resource availability and competition shape the evolution of survival and growth ability in a bacterial community." PLoS One 8.9 (2013): e76471."

Thank you for suggesting this reference. Upon reading it we realized that it fits nicely in the paper and we now cite it.

6) Please check the Fig S1 legend "??"

Thanks for noticing, we have fixed this.

Reviewer #3 (Comments for the Author):

In this paper, Dos Santos et al. attempt to classify interactions between four species in a synthetic microbial community by growing each species in the spent media of each other species. However, they find that the outcomes of these measurements do not conform with the expectations of a model. In particular, they find that one species (Ct) has a pattern of interactions such that it grows best in the pure spent media of two of the other species, and slightly worse when the spent media is supplemented by more of the original base media. They determine that this is due to cross-feeding of particular metabolites by Ct.

The work appears thorough and carefully done, and the paper is well-written overall (with a few key areas to improve clarity; see below). However, I am struggling to see how their results have general significance for the field, rather than being just a quirk of their particular system. To the extent the authors claim a general significance for these results, I find their arguments misleading.

The authors interpret their results as being an important example of how "cross-feeding could alleviate the negative effects of a challenging environment" (lines 123-124), but I find this claim ill-defined. The essence of the claim appears to be that the authors believe there is a fundamental difference between the "benign environment" scenarios in the first row of Fig. S1, and the "inhibitory environment" scenarios in the second row. The difference in every case is just the presence of the

inhibitor C3, but how is the presence of this inhibitor any different from simply assigning a slower growth rate to species S2 in the environment? If species S1 actively removed C3, then I can see how it plays a dynamic role in the different interaction architectures, but as is, it just seems like a fixed part of the background medium that doesn't need any explicit modeling (outside of the $r_{\{i,j\}}$ parameters in Eq. 1). Related to this, the authors use the word "inhibitory" for this effect (e.g., in the abstract on line 19), but my point is that inhibition must be relative to something, and it's not clear what, since the authors are only considering a single medium type.

We apologize if our logic was not clear. We are arguing that experimental design of spent medium assays can be hard to interpret: Typically, one expects that adding back the carbon sources into a spent medium would restore growth to at least the original level. "Inhibitory" refers to the fact that for that one species, replenishing carbon sources results in very poor growth relative to the original medium that has at least the same concentration of carbon sources. Our point then is that this makes spent medium experiments difficult to interpret, as there is an added layer of complexity, which is why we provide a model and guidelines to help interpreting similar experiments.

Related to this, I didn't understand the significance of the inhibitory components of M9 to the later discovery that Ct was consuming metabolites produced by At and Ms. It doesn't seem very surprising that they can alter various components of the medium and change the lag time of a population, but I don't see what this has to do with the cross-feeding story that arises later. (The authors claim that the "minimal medium delays Ct's growth" (lines 278-279), but again, delay relative to what?) To me, the point here isn't that M9 delays growth, but simply that the metabolites from At and Ms accelerate growth relative to the base media. The inhibitory effects of the M9 components seem completely irrelevant to this, outside of the authors' speculation that the cross-fed metabolites might alleviate this stress (lines 384-385, but there is no evidence for this).

From an experimental design perspective, we are arguing that it is quite tricky to detect cross-feeding. We designed several spent media assays that would help us to recognize it, but its signal was obscured because replenishing the medium had this strange effect that it prolonged the lag phase (relative to the original growth medium) where it was not expected. We then proceeded to disentangle these effects to allow us to correctly classify the interactions. The main point of the paper is not to highlight the cross-feeding in itself but more to discuss our approaches to recognizing it in other contexts. The take-home message then is: if you carry out spent medium experiments and observe similarly strange effects, this might be due to growth being slowed down by the higher concentration of certain compounds in the medium. We expect that this will occur often in the lab, as we know that it is difficult to find optimal growth conditions for many species. We hope this is now clearer.

Line-by-line comments

- Lines 87-91: Here the authors introduce their system, but I think they need to provide a little more background on where this system comes from and why it is interesting beyond just citing their previous work.

We have now expanded that section to highlight why we are following up on our previous work: "Here, we aimed to decipher the interactions in a synthetic community we studied previously, composed of four bacterial species: *Agrobacterium tumefaciens* (At), *Comamonas testosteroni* (Ct), *Microbacterium saperdae* (Ms), and *Ochrobactrum anthropi*

(Oa) that can grow and degrade industrial machine oils (31). This community was dominated by positive interactions when we compared their growth in mono- and co-cultures in the oil. The chemical complexity of the growth medium and the use of co-cultures made it difficult for us to understand the mechanisms behind these positive interactions. Here, we sought to provide a more controlled environment and used a defined minimal medium (MM) to study the mechanisms behind the interactions between the four species.”

- Lines 126-127: The title of this subsection sounds too vague. What are the "others"? What does "following anticipated scenarios" mean? I would suggest something like "At and Oa respond to other species according to model-predicted classifications."

Thank you for the suggestion! We changed it to: "At and Ct responded to other species according to model predictions"

- Lines 144-159: The link between the experimental media conditions (conditions I-V) and the theoretical interaction classes (EC, EC+IC, etc.) was confusing. Eventually I discovered Fig. S12 that seems to be the precise definition of the correspondence (is this what they use to construct Figs. 1A,B?), but this wasn't mentioned until the Discussion. The authors need to explain these details more precisely in this section, because they are crucial for everything that follows.

Indeed, it seems (now) Fig. S1B was a bit too buried. We now mention it in the caption of Fig. 1 "(with the help of the decision tree in Fig. S1B)", and the main text "with the help of a decision tree we constructed based on the predictions of the model (see Fig. S1B)" when we first explain how we classify interactions.

- Line 145: There should be a reference around here to the mathematical model details in Supplementary Note S3.

Done.

- Fig. 1D: The lines here are hard to distinguish. My understanding is that for conditions III-V there are supposed to be two different colors for each line style, but the legend confuses this because it also shows these lines in a third color (brown) which doesn't seem to actually be used in the plot. I think it would be best to just enumerate each line style and color combination separately in the legend, plus use a better set of line styles (the line styles for conditions III and V are too similar).

Yes, this was pointed out by all reviewers, and we have now clarified the legend and changed the colors for better distinction.

- Fig. 1D: Why don't the authors use the log-scale version of this plot (Fig. S11) here instead? Their claim is about lag time, but lag time differences are practically impossible to separate from growth rate differences on linear-scale plots, so the log-scale is really essential here. In any case, whichever alternative version of this plot ends up in the supplement should be cited in the caption and accompanying main text.

After some consideration, we decided to change all the OD plots in the manuscript to log scale. We moved the linear scale version of the plot to the supplement. We now agree with the reviewer that this is the better way to show the data.

- Fig. 2: If the point of this figure is to show how the lag times differ across media conditions, why didn't the authors plot exactly that? I do appreciate the authors' inclusion of so many raw growth

curves, so that readers can see the raw data as much as possible, but for the main text I think the paper needs more summary figures that more directly show the important trends (e.g., in lag time specifically, not necessarily the whole growth curve). Distinguishing the growth curves was especially hard in these plots, since the color schemes are not so obvious and some trends are even non-monotonic (as in Fig. 2A). I would certainly still want to include the growth curves themselves, but they could be supplementary figures. (It would also be more helpful to plot these growth curves on log-scales if the key feature is lag time differences.) This comment applies to Fig. 3B as well.

We understand the concerns with Fig. 2 (and Fig. 3B). All figures are now in log scale. In order to make the different growth curves easier to read (Fig. 2A, B and D), we added shapes for each condition in addition to the gradient of gray/black. In Fig. 3B we also added the lag phase thresholds that were used to perform statistics on the growth curves of Fig. 3B to help the reader see the shift in the length of the lag phase. However, we decided that now the growth curves show quite clearly the trends described in the result sections concerning the length of the lag phase. We feel that the raw growth curves display details (like the bump described in Fig. 4F) that would be lost in summary figures. We have therefore decided not to replace them with summary figures.

- Line 361: There is a glitch in one of the citations here.

Thank you for pointing this out. It is now corrected.

- Fig. S1 caption: There is a glitch in a reference to what probably should be Supplementary Note S3.

Fixed, thanks for noticing!

- Fig. S4: The significance markers don't make sense in some of the cases here. Aren't they testing for significant differences of cases III-V compared to case II (set to 1)? Some data that almost exactly matches case II (e.g., At grown on At, case V) is still shown as significant. Shouldn't that be not significant?

Thank you for pointing this out. We redid the statistics on normalized data, and the transformed data and the statistics are displayed in (now) Fig. S3B. The significance values seem to be more intuitive now.

- Fig. S5: Why don't the authors directly plot OD vs. CFUs/mL, rather than plotting each separately vs. time? That would show the correlation (or not) more clearly, especially since the axes for the two here are on completely different scales (OD is linear, CFUs/mL is log, so comparing them is very hard).

To help the comparison between the CFUs/mL and the OD data, we now put all the graphs in a log₁₀ scale. We however do prefer having the curves in separate graphs so that the data concerning the different species does not overlap too much (we want to avoid graphs that are too busy).

- Tables S1-S8: These tables contain information about the comparisons of growth curve traits (lag times and final yields), but these quantities themselves are not plotted anywhere (related to aforementioned comment on Figs. 2 and 3B). These plots seem critical.

In Fig. 3B, we added the lag phase thresholds that were used to compute the statistics to help the reader see the shift in the length of the lag phase. In addition, as explained in a previous comment, we considered making summary graphs plotting the final yield and the lag phase. However, we believe that the growth curves clearly show the trends that we described in the results section and that it would only add more graphs in the paper without necessarily improving the understandability of our data.

- Tables S1-S8: Also, the supplement would be easier to navigate if the tables were all separate from the figures, rather than intermixed.

We have now separated the figures and tables from one another.

- Fig. S11: There is an arrow in this figure that I believe is explained by the caption to Fig. 4, but it should be explained in this figure's caption as well.

Done!

- Eq. 1 (Supplementary Note S3): In Eq. 1b, ρ_i should not be written as a function of C_k , because the index k is summed over in the denominator of the lag time factor. This dependence should also be removed in Eqs. 1a and 1c, where it suggests that the growth rate of species i depends on pairs of compounds, whereas it only depends on each compound separately with a global lag time factor. Also, the authors should cite previous work that developed this model, or explain which parts of the model are original to this paper.

This was an error, of course. Thank you for reading the manuscript carefully enough to spot it!

August 2, 2022

Dr. Sara Mitri
University of Lausanne
Lausanne
Switzerland

Re: mSystems00239-22R1 (Classifying interactions in a synthetic bacterial community is hindered by inhibitory growth medium)

Dear Dr. Sara Mitri:

Thank you for submitting your manuscript to mSystems. There are still a few issues to address before it can be accepted for publication. The most important one is the question raised by the 2nd reviewer, namely how to differentiate between compounds increasing growth directly (cross-feeding) and compounds increasing growth indirectly by alleviating negative effects of the medium (cross-detoxification). In my opinion, you have answered that point, but given the reviewer comments, a clarification is a good idea and will help to better emphasize the main message of this manuscript. Please also note that figure captions and figures were missing in the merged pdf and that the link to the model implementation was not active (please provide the full url). Also, it would be helpful for interested readers if data behind figures were available, e.g. shared on DataDryad or an accompanying web page.

Below you will find instructions from the mSystems editorial office and comments generated during the review.

Preparing Revision Guidelines

Sincerely,

Karoline Faust

Editor, mSystems

Journals Department
American Society for Microbiology

Reviewer comments:

Reviewer #1 (Comments for the Author):

The authors made some improvements to their manuscript, in particular they explored the parameter-dependence of their model more thoroughly. However, unfortunately I feel that my questions 1) and 2) were not really answered or I disagree with the answers. I don't want to delay publication further and at some points we may just end up agreeing that we disagree. But, I feel this concerns should be (in the interest of the authors) at least carefully thought about and - even better - addressed in the manuscript.

1) I am not concerned about the definition of the term fitness or its measurement (which is what the authors address in their response), but about the issue that what is measured is not an interaction. Interaction means that one species impacts (!) the growth/fitness of another species. A sheer change in lag time because of a change in conditions is not an interaction because this change in lag time does not (necessarily) come from species interactions - e.g. a change in lag time may change fitness but does not inform about interactions. Accordingly, I find it a strong (and honestly wrong) statement to write in the manuscript that media impacted interactions - which already the title implies - when the measurements are not really addressing interactions.

I also disagree that steady state population densities are only meaningful in chemostats. There are many papers that use batch culture with daily dilution and measurement of population densities before dilution. These systems reach steady states. Moreover, it is shown that for Lotka Volterra models constant and daily dilutions lead to same outcomes¹, accordingly they also can be used in such cases.

2) I also slightly disagree that the inability to distinguish dilution from addition effects upon replenishing media is "inherent to the design of spent media experiments". A simple way to at least lower the first and check the second is to replenish with small volumes of concentrated solutions. This allows to replenish substances with minimal volume change.

Reviewer #3 (Comments for the Author):

I thank the authors for their many revisions that have improved the clarity of the paper. However, I feel they have not fully addressed the conceptual concern I have, which regards the interpretation of how the abiotic medium and cross-feeding interactions jointly affect the growth of a species.

Their responses to my original comments on this topic suggest that they believe the main message of the paper to be that interpreting spent-media assays is difficult. I certainly agree with this point and think that their experiments are a good demonstration of it. But the paper text itself presents a much more specific interpretation of their data, that "even simple, defined growth media can have inhibitory effects on some species and that such negative effects need to be included in our models" (lines 18-21) and "cross-feeding could alleviate the negative effects of a challenging environment" (lines 18-21). They prominently make these statements in the abstract, introduction, and throughout the main body, so it seems that they consider this a major result of the paper, rather than the paper just being a cautionary tale about spent-media assays. So I would like the authors to defend this specific interpretation.

In my view the issue is whether there is any difference between 1) a situation where the abiotic medium alone inhibits growth but another strain produces something that alleviates that inhibition (their current interpretation in the paper), and 2) a situation where the baseline growth is whatever the abiotic medium allows and the other strain produces something that accelerates growth. Do the authors consider these equivalent, biologically as well as mathematically?

If they don't, then I would like them to precisely articulate what the difference is and how it would manifest in the model and experiment, because I don't understand it.

If they do, then I think they should expand their explanation of the results to explain that these are equivalent interpretations, and can be realized in the model accordingly. I acknowledge this may sound like nit-picking, but I think it is important because a major driver of their results seems to be the apparent mismatch between their model predictions (Fig. 1A) and the spent-media assays in the case of Ct growing on spent media from At and Ms (Fig. 1B), specifically in the cases of SM and SM/2 + MM. They present this as if the models are fundamentally missing something (the inhibition of MM), but it seems to me that the effect of At and Ms on Ct should still fall under the category of niche separation (since they don't appear to overlap in carbon sources) + cross-feeding. The difference between the model prediction for this case and the experiment is that their model predicts little difference between SM and SM/2 + MM, whereas the experiments show that SM/2 + MM results in much less growth of Ct compared to SM alone. But isn't this a matter of how strongly the cross-feeding benefits Ct? What if the cross-fed metabolites

from At and Ms have a really strong effect on Ct's growth, relative to Ct's growth in the basal media? Then I would expect Ct in SM alone to significantly outperform itself in SM/2 + MM, since the latter case has only half as much of the beneficial cross-fed metabolites. And I would still count this as the niche separation + cross-feeding case, not something fundamentally different. Indeed, they appear to see this in some parameter cases in Fig. S8.

If this is the case, then I think their presentation of the model/experiment mismatch is misleading, because the mismatch is not inherent to the interaction cases they enumerate, but rather it is because of the particular parameters they chose to realize those cases in the model. In particular, it contradicts their conclusion that "negative effects need to be included in our models," because the model already covers the case they observe.

Response to referees

Manuscript number: mSystems00239-22R1

Title: "Classifying interactions in a synthetic bacterial community is hindered by inhibitory growth medium"

We would like to thank the two reviewers for their consideration of our revised manuscript and their detailed comments. We hope that this time we have better addressed the remaining concerns. Reviewers' comments are written in black below, and our responses in blue font and indented text. We hope that you will find that our updated manuscript now merits publication.

Reviewer #1 (Comments for the Author):

The authors made some improvements to their manuscript, in particular they explored the parameter-dependence of their model more thoroughly. However, unfortunately I feel that my questions 1) and 2) were not really answered or I disagree with the answers. I don't want to delay publication further and at some points we may just end up agreeing that we disagree. But, I feel this concerns should be (in the interest of the authors) at least carefully thought about and - even better - addressed in the manuscript.

Thank you! We will do our best to clarify these two points.

1) I am not concerned about the definition of the term fitness or its measurement (which is what the authors address in their response), but about the issue that what is measured is not an interaction. Interaction means that one species impacts (!) the growth/fitness of another species. A sheer change in lag time because of a change in conditions is not an interaction because this change in lag time does not (necessarily) come from species interactions - e.g. a change in lag time may change fitness but does not inform about interactions. Accordingly, I find it a strong (and honestly wrong) statement to write in the manuscript that media impacted interactions - which already the title implies - when the measurements are not really addressing interactions.

If we understand this correctly, the reviewer disagrees with calling an indirect, environmentally-mediated effect of one species on another an interaction. As we include such effects in our definition of interactions, we now spell this out explicitly at the beginning of the introduction. We also cite other literature that uses this definition. "This affects other organisms living in their proximity, resulting in "indirect" or "environmentally mediated" interactions (15, 36)."

I also disagree that steady state population densities are only meaningful in chemostats. There are many papers that use batch culture with daily dilution and measurement of population densities before dilution. These systems reach steady states. Moreover, it is shown that for Lotka Volterra models constant and daily dilutions lead to same outcomes¹, accordingly they also can be used in such cases.

Perhaps we were not precise enough. We see transfer experiments (batch cultures with repeated dilutions) to be very comparable to chemostat experiments, and we agree that these can also reach steady states.

2) I also slightly disagree that the inability to distinguish dilution from addition effects upon replenishing media is "inherent to the design of spent media experiments". A simple way to at least lower the first and check the second is to replenish with small volumes of concentrated solutions. This allows to replenish substances with minimal volume change.

We agree that replenishing with highly concentrated solutions can diminish dilution effects, but they cannot be completely eliminated. By combining experiments where we dilute the SM or not, our experimental design tries to distinguish these effects. They are just difficult to distinguish with a single experimental condition for the SM.

Reviewer #3 (Comments for the Author):

I thank the authors for their many revisions that have improved the clarity of the paper. However, I feel they have not fully addressed the conceptual concern I have, which regards the interpretation of how the abiotic medium and cross-feeding interactions jointly affect the growth of a species.

Thank you! We have tried to make our interpretation more explicit, as detailed below.

Their responses to my original comments on this topic suggest that they believe the main message of the paper to be that interpreting spent-media assays is difficult. I certainly agree with this point and think that their experiments are a good demonstration of it. But the paper text itself presents a much more specific interpretation of their data, that "even simple, defined growth media can have inhibitory effects on some species and that such negative effects need to be included in our models" (lines 18-21) and "cross-feeding could alleviate the negative effects of a challenging environment" (lines 18-21). They prominently make these statements in the abstract, introduction, and throughout the main body, so it seems that they consider this a major result of the paper, rather than the paper just being a cautionary tale about spent-media assays. So I would like the authors to defend this specific interpretation.

As we see it, our paper provides an example of why interpreting spent-media assays is difficult. This difficulty comes partly because of the second point: when growth media are inhibitory, the results of spent media assays may look surprising (e.g. a species grows really badly in its own replenished spent medium). These two points are thus related, and both part of our cautionary message. The third point the reviewer mentions is that if positive effects are observed in an inhibitory medium, it is not necessarily because one species alleviates these inhibitory effects, but it can be another mechanism: crossfeeding. Being able to model these two mechanisms allowed us to distinguish them mathematically and then look for their signatures in our data, which matched what we found with our molecular analysis,

All of this boils down to how to interpret the data from these experiments and classify interactions. We have tried to clarify all of this further in the manuscript. For example, the last sentence of the introduction now reads: "Our findings show that pinpointing the nature of positive interactions can be quite challenging -- because growth media can sometimes be inhibitory and because inhibitory effects can be alleviated by either cross-detoxification or cross-feeding --, but that spent media assays and growth curve measurements can nevertheless be sufficient to distinguish between cross-feeding and cross-detoxification." To the discussion, we have also added the following sentence: "These features [shown in Fig. 4D, E and F], obtained by high resolution growth curve measurement until stationary phase, make it possible to distinguish between cross-feeding and cross-detoxification without the need for further molecular analyses." and in the final paragraph "High resolution growth curve measurements can even help to distinguish cross-feeding from cross-detoxification."

In my view the issue is whether there is any difference between 1) a situation where the abiotic medium alone inhibits growth but another strain produces something that alleviates that inhibition (their current

interpretation in the paper), and 2) a situation where the baseline growth is whatever the abiotic medium allows and the other strain produces something that accelerates growth. Do the authors consider these equivalent, biologically as well as mathematically?

There is definitely a difference between these two situations. They are not equivalent, neither biologically, nor mathematically.

If they don't, then I would like them to precisely articulate what the difference is and how it would manifest in the model and experiment, because I don't understand it.

We illustrate the difference in Fig. 4: the mathematical difference is shown in panels D and E, where the final population size differs and there is a bump in the beginning of the growth curve. We have highlighted these differences with arrows. We then show in panel F that our experiments are more consistent with the model in Fig. 4D. We have tried to articulate the difference better in the text (see discussion excerpt above).

If they do, then I think they should expand their explanation of the results to explain that these are equivalent interpretations, and can be realized in the model accordingly. I acknowledge this may sound like nit-picking, but I think it is important because a major driver of their results seems to be the apparent mismatch between their model predictions (Fig. 1A) and the spent-media assays in the case of Ct growing on spent media from At and Ms (Fig. 1B), specifically in the cases of SM and SM/2 + MM. They present this as if the models are fundamentally missing something (the inhibition of MM), but it seems to me that the effect of At and Ms on Ct should still fall under the category of niche separation (since they don't appear to overlap in carbon sources) + cross-feeding. The difference between the model prediction for this case and the experiment is that their model predicts little difference between SM and SM/2 + MM, whereas the experiments show that SM/2 + MM results in much less growth of Ct compared to SM alone. But isn't this a matter of how strongly the cross-feeding benefits Ct? What if the cross-fed metabolites from At and Ms have a really strong effect on Ct's growth, relative to Ct's growth in the basal media? Then I would expect Ct in SM alone to significantly outperform itself in SM/2 + MM, since the latter case has only half as much of the beneficial cross-fed metabolites. And I would still count this as the niche separation + cross-feeding case, not something fundamentally different. Indeed, they appear to see this in some parameter cases in Fig. S8.

The initial mismatch was because our model was too simple, as it lacked the inhibiting effect of NC and MM on Ct. Once we updated the model the mismatch disappeared and as the reviewer points out, Ct grows much better in SM compared to SM/2+MM, partly because the positive effects of cross-feeding are diluted and because of the inhibitory effect of adding MM. The model doesn't capture this quantitatively because we weren't fitting parameters but rather looking at qualitative effects. So indeed, cross-feeding likely has a stronger effect in reality than the parameter we chose in our model. In sum, it is niche separation + cross-feeding, just in an inhibitory medium (as shown in Fig. 4C). So the second model that includes inhibitory effects qualitatively captures the experimental data.

If this is the case, then I think their presentation of the model/experiment mismatch is misleading, because the mismatch is not inherent to the interaction cases they enumerate, but rather it is because of the particular parameters they chose to realize those cases in the model. In particular, it contradicts their conclusion that "negative effects need to be included in our models," because the model already covers the case they observe.

The first model (with no inhibitory medium) does cover the case we observe, but the prediction does not at all match the observation. Only the second model (with inhibitory medium) matches the observation. Hence the need for the second model. We have tried to clarify all of this better in the manuscript, particularly by updating the caption of Fig. 4, which was not very clear before.

September 15, 2022

Dr. Sara Mitri
University of Lausanne
Lausanne
Switzerland

Re: mSystems00239-22R2 (Classifying interactions in a synthetic bacterial community is hindered by inhibitory growth medium)

Dear Dr. Sara Mitri:

There are still unresolved concerns by the third reviewer. This work claims that cross-feeding versus cross-detoxification can be distinguished based on characteristics of the growth curve in spent medium assays (higher yield in cross-feeding). The reviewer is mainly concerned that a quantitative model differentiating between these cases will not always be able to do so, depending on parameter choice. However, the qualitative differences between those two cases deserve to be discussed and tested in the research community.

Thus, your manuscript has been accepted, and I am forwarding it to the ASM Journals Department for publication. For your reference, ASM Journals' address is given below. Before it can be scheduled for publication, your manuscript will be checked by the mSystems production staff to make sure that all elements meet the technical requirements for publication. They will contact you if anything needs to be revised before copyediting and production can begin. Otherwise, you will be notified when your proofs are ready to be viewed.

Publication Fees:

If you would like to submit a potential Featured Image, please email a file and a short legend to msystems@asmusa.org. Please note that we can only consider images that (i) the authors created or own and (ii) have not been previously published. By submitting, you agree that the image can be used under the same terms as the published article. File requirements: square dimensions (4" x 4"), 300 dpi resolution, RGB colorspace, TIF file format.

We recognize that the video files can become quite large, and so to avoid quality loss ASM suggests sending the video file via <https://www.wetransfer.com/>. When you have a final version of the video and the still ready to share, please send it to mSystems staff at msystems@asmusa.org.

Sincerely,

Karoline Faust
Editor, mSystems

Journals Department
Fig. S3: Accept
Fig. S8: Accept
Fig. S6: Accept
Fig. S5: Accept
Fig. S2: Accept
Note S1: Accept
Fig. S4: Accept
Table S1: Accept
Fig. S7: Accept
Fig. S1: Accept